# LABEL-ENCODING RISK MINIMIZATION UNDER LABEL INSUFFICIENT SCENARIOS

## ABSTRACT

The Empirical Risk Minimization (ERM) adopts the supervision information, *i.e.*, class labels, to guide the learning of labeled samples and achieves great success in many applications. However, many real-world applications usually face the label insufficient scenario, where there exist limited or even no labeled samples but abundant unlabeled samples. Under those scenarios, the ERM cannot be directly applied to tackle them. To alleviate this issue, we propose a Label-encoding Risk Minimization (LRM), which draws inspiration from the phenomenon of neural collapse. Specifically, the proposed LRM first estimates the label encodings through prediction means for unlabeled samples and then aligns them with their corresponding ground-truth label encodings. As a result, the LRM takes both the prediction discriminability and diversity into account and can be utilized as a plugin in existing models to address scenarios with insufficient labels. Theoretically, we analyze the relationship between the LRM and ERM. Empirically, we demonstrate the superiority of the LRM under several label insufficient scenarios, including semi-supervised learning, unsupervised domain adaptation, and semi-supervised heterogeneous domain adaptation. The code will be released soon.

## 1 INTRODUCTION

With abundant high-quality human-annotated samples, deep learning techniques have achieved remarkable advances in various applications (Shi et al., 2021; Subramanian et al., 2022; Wang et al., 2023). One key principle behind their success is the Empirical Risk Minimization (ERM), which adopts the supervision information, *i.e.*, class labels, to guide the learning of labeled samples. In practice, however, we often encounter some label insufficient scenarios (Cui et al., 2020), where the labeled samples are limited or may even be absent altogether. For the former, we can utilize a large number of unlabeled samples to assist the learning of labeled samples, which falls within the scope of semi-supervised learning (Sohn et al., 2020; Zhang et al., 2021a; Chen et al., 2022). On the contrary, as for the latter, a popular solution is to borrow the knowledge from a similar and label sufficient domain, *i.e.*, source domain, for facilitating the learning of unlabeled samples, which pertains to the field of transfer learning (Pan & Yang, 2010; Yang et al., 2020). The commonality of those techniques is to fully utilize unlabeled samples for improving the generalization capability in label insufficient scenarios. These scenarios are not uncommon in practical applications. For instance, in the field of industrial defect detection, the annotation of defective images demands a substantial number of professionals, which is expensive and time-consuming (Zhang et al., 2023b). As the ERM heavily relies on the guidance of the label information and fails to fully utilize the potential of unlabeled samples, leading to suboptimal performance in label insufficient scenarios. Hence, this paper focuses on mining the potential of unlabeled samples to address the issue of insufficient labeling.

To achieve this, a simple and popular approach is the entropy minimization (EntMin) (Grandvalet & Bengio, 2004). It can be regarded as a direct extension of ERM to unlabeled samples. Specifically, it utilizes the soft-labels of unlabeled samples, which are assigned by the classifier during the training process, for guiding their own learning, and it pushes samples far from the decision boundary. One problem of EntMin is that the soft-labels assigned by the classifier could be mainly from dominant categories with large numbers of samples, resulting in a decrease in prediction diversity (Cui et al., 2020) that the samples are prone to be pushed towards the majority categories. One reason for that lies in the absence of more appropriate guidance information for unlabeled samples. So we want to ask "*for unlabeled samples, is there more precise guidance information available?*"

To seek a potential solution to the above problem, we delve deeper into the essence of the ERM. A recently discovered phenomenon of the neural collapse (Papyan et al., 2020) has captured our attention, which prevalently exists in trained neural networks under the ERM framework. One property of neural collapse is the neural class-mean collapse (NCC), which reveals that the learned features[1] of samples from a class collapse to their corresponding class mean. We prove that the underlying cause of NCC roots in the use of *label encoding*, *i.e.*, one-hot label encoding, which leads to the collapse of the features through back-propagation. That is, all samples associated with the sample category need to be mapped to a label encoding corresponding to that category. In other words, *the guidance information of the ERM for the samples in a specific category is the corresponding label encoding*. Moreover, under label insufficient scenarios studied in this paper, the label encodings of labeled samples remain consistent with those of unlabeled samples. Accordingly, it is reasonable to apply label encodings as guidance information to learn from unlabeled samples. Inspired by this, we propose the Label-encoding Risk Minimization (LRM), a generalization of ERM, to handle unlabeled samples. Specifically, the proposed LRM first estimates the label encodings based on unlabeled samples by calculating the prediction mean in each class. Then, the LRM minimizes the label-encoding risk, *i.e.*, the discrepancy between the estimated label encodings and their corresponding label encodings. Since these label encodings serve as accurate supervision information, in conjunction with many existing models, the LRM can enhance their generalization performance under label insufficient scenarios.

The contributions of this paper are three-fold. (1) We theoretically reveal the primary rationale behind the NCC, that is, the use of label encoding results in the collapse of features through back-propagation. (2) We propose LRM, a refined generalization of the ERM to unlabeled samples. Also, we theoretically analyze the relationship between the LRM and ERM. (3) Extensive experiments conducted under several label insufficient scenarios, including semi-supervised learning (SSL), unsupervised domain adaptation (UDA), and semi-supervised heterogeneous domain adaptation (SHDA), verify the effectiveness of the proposed LRM.

## 2 RELATED WORK

**Label Insufficient Scenarios**. In this paper, we mainly focus on three learning tasks under label insufficient scenarios, *i.e.*, SSL (Sohn et al., 2020; Zhang et al., 2021a; Chen et al., 2022), UDA (Ganin et al., 2016; Long et al., 2018; Xu et al., 2019; Rangwani et al., 2022; Zhang et al., 2023a), and SHDA (Yao et al., 2019; Li et al., 2020; Gu et al., 2022; Fang et al., 2022). Specifically, SSL leverages limited labeled samples and massive unlabeled samples to improve the generalization capability. For example, FlexMatch (Zhang et al., 2021a) utilizes curriculum pseudo learning for enhancing the performance of SSL, and DST (Chen et al., 2022) mitigates the impact of incorrect pseudo-labels during the iterative self-training process. On the other hand, UDA aims to improve the learning of a target domain with unlabeled samples by harnessing the knowledge from a source domain with sufficient labeled samples. For instance, DANN (Ganin et al., 2016) and CDAN (Long et al., 2018) bridge the source and target domains through adversarial training. AFN (Xu et al., 2019) progressively adapts the feature norms of the two domains to a broad range of values. Recently, SDAT (Rangwani et al., 2022) enhances the stability of domain adversarial training and seeks a flat minimum. Considering the heterogeneity of the features between domains, SHDA utilizes additional limited labeled samples in the target domain to improve the transfer effect. As an example, STN (Yao et al., 2019) adopts the soft-labels of unlabeled target samples to align the conditional distributions across domains. Another example is KPG (Gu et al., 2022), which leverages key samples to guide the matching process in optimal transport. Among those learning tasks, to boost performance, it is vital to fully utilize the information of unlabeled samples. EntMin (Grandvalet & Bengio, 2004) is commonly used to leverage unlabeled samples to improve prediction discriminability. For example, in SSL, Berthelot et al. (2019) utilizes EntMin to estimate low-entropy labels for data-augmented unlabeled samples. Moreover, Yin et al. (2022) adopts the entropy of the predicted distribution as a confidence measure to filter pseudo-labels effectively. In UDA, EntMin is utilized in (Long et al., 2016; Vu et al., 2019) to obtain a more reliable decision boundary in the unlabeled target domain. In addition, Batch Nuclear-norm Maximization (BNM) (Cui et al., 2020) is proposed to maximize the nuclear norm of the prediction matrix derived from unlabeled samples, which ensures both the prediction discriminability and diversity. Similar to LRM, EntMin and BNM can also be seamlessly

---

[1]In this paper, the features default to the output of the penultimate layer in a deep neural network.

embedded into any existing SSL, UDA, and SHDA approaches. However, unlike them, LRM adopts label encodings as the supervision information, which achieves both the prediction discriminability and diversity in a classification fashion.

**Neural Collapse**. Neural collapse is initially observed in (Papyan et al., 2020), which unveils that the features and classifier weights exhibit remarkably elegant properties after sufficient training. Subsequently, some theoretical studies (Kothapalli, 2023) mainly pursue two lines in an attempt to uncover the rationale behind neural collapse. One line of studies is based on the principle of unconstrained features (Mixon et al., 2022; Fang et al., 2021; Zhou et al., 2022), which assumes that the features can be freely optimized. On the contrary, another line of studies is grounded in the principle of local elasticity (He & Su, 2020; Zhang et al., 2021b), which strives to capture the progressive differentiation of class features. However, different from them, our work primarily focuses on the NCC property, *i.e.*, the learned features of within-class samples collapse to the corresponding class mean. We not only theoretically analyze the key rationale behind the NCC, but also derive the label-encoding risk to guide the learning of unlabeled samples.

## 3 EMPIRICAL RISK MINIMIZATION

The ERM aims to provide theoretical guidance for the learning of labeled samples, which can be derived from the expected risk minimization. Under the expected risk minimization framework, we are interested in learning a function $f(\cdot)$ from a set of potential function sets $\mathcal{F}$. Such function $f(\cdot)$ characterizes the connection between a random feature vector $\mathbf{x}$ and the corresponding target vector $\mathbf{y}$. For this purpose, we need a non-negative real-valued loss function $\mathcal{L}(\cdot, \cdot)$, which tells us how much it "hurts" to make the prediction $f(\mathbf{x})$ when the actual target $\mathbf{y}$, for a sample $(\mathbf{x}, \mathbf{y}) \sim P$. Then, the expected risk is defined as the expectation of the loss function $\mathcal{L}(\cdot, \cdot)$ over the distribution $P$ as

$$\mathcal{R}_{exp}(f) = \mathbb{E}_P[\mathcal{L}(f(\mathbf{x}), \mathbf{y})] = \int \mathcal{L}(f(\mathbf{x}), \mathbf{y}) \mathrm{d}P(\mathbf{x}, \mathbf{y}). \tag{1}$$

An inevitable obstacle of the above calculation is that it is impossible to know the distribution $P$ in most realistic scenarios. To tackle with it, we can use an approximation, *i.e.*, the empirical risk, by averaging the loss function $\mathcal{L}(\cdot, \cdot)$ over a set of training samples $\mathcal{D} = \{(\mathbf{x}_i, \mathbf{y}_i)\}_{i=1}^n$, where $(\mathbf{x}_i, \mathbf{y}_i) \sim P$ for $i = 1, ..., n$. Accordingly, the empirical risk can be formulated as

$$\mathcal{R}_{emp}(f) = \frac{1}{n} \sum_{i=1}^n \mathcal{L}(f(\mathbf{x}_i), \mathbf{y}_i). \tag{2}$$

Finally, we learn function $f^*$ that minimizes Eq. (2), which is known as ERM:

$$f^* = \arg\min_f \mathcal{R}_{emp}(f) = \arg\min_f \frac{1}{n} \sum_{i=1}^n \mathcal{L}(f(\mathbf{x}_i), \mathbf{y}_i). \tag{3}$$

## 4 LABEL-ENCODING RISK MINIMIZATION

The ERM has been successfully applied to various label sufficient scenarios. However, it is not so powerful under label insufficient scenarios. In the following sections, we commence by introducing the NCC property, which reveals that in the context of classification, the essence of ERM lies in mapping all the within-class samples to the corresponding ground-truth label encoding. Inspired by this, we propose the LRM, a refined generalization of the ERM to unlabeled samples, which can be utilized to handle challenging label insufficient scenarios. Then, we establish connections between LRM and NCC as well as ERM, and also compare LRM with EntMin. Finally, we detail how to concretely apply the LRM to label insufficient scenarios.

### 4.1 AN ANALYSIS ON NEURAL CLASS-MEAN COLLAPSE

Recently, under the ERM framework, a widespread phenomenon called neural collapse has been discovered in (Papyan et al., 2020). One of its attractive properties is the NCC, where the optimized features in the same class collapse to the corresponding class mean. We delve deeper into the cause of NCC and find that it arises from the use of label encoding, where the labels of within-class samples

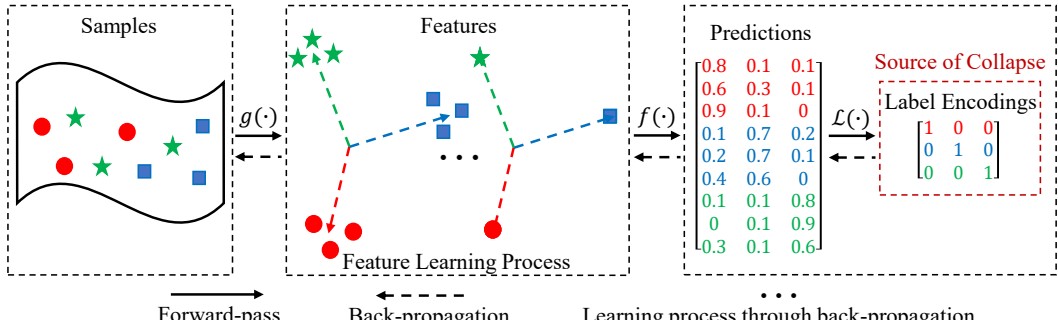

Figure 1: An intuitive explanation for the NCC. We can observe that the nine samples are mapped into three label encodings, resulting in the collapse of features through back-propagation.

are all encoded to be identical. *In this paper, label encoding refers to one-hot encoding by default.* Accordingly, we infer that the key rationale behind NCC is the use of label encoding causes the collapse of the features via back-propagation. An intuitive explanation is depicted in Figure 1. There are nine samples belonging to three distinct categories. However, there are only three label encodings, *i.e.*, $[1, 0, 0]$, $[0, 1, 0]$, and $[0, 0, 1]$. Hence, the task of classification is actually to map nine samples into three label encodings according to their categories. This forces the optimized features of nine samples to collapse into three class means. In the following theorem, we prove our inference based on the aforementioned principle of unconstrained features.

**Theorem 1** *Given an injective function $f(\cdot)$: $\mathbb{R}^d \to \mathbb{R}^C$ and a loss function $\mathcal{L}(\cdot, \cdot)$: $\mathbb{R}^C \times \mathbb{R}^C \to [0, +\infty)$ satisfying that $\mathcal{L}(y_1, y_2) = 0$ indicates $y_1 = y_2$, then for any feature matrix $\mathbf{H} \in \mathbb{R}^{n \times d}$ and its one-hot label matrix $\mathbf{Y} \in \mathbb{R}^{n \times c}$, when $\mathcal{L}(f(\mathbf{H}), \mathbf{Y})$ w.r.t. $\mathbf{H}$ and $f$ is minimized to 0, we have*

$$\mathbf{h}_{1,c}^* = \cdots = \mathbf{h}_{n_c,c}^* = \frac{1}{n_c} \sum_{i=1}^{n_c} \mathbf{h}_{i,c}^*, \forall c \in \{1, \ldots, C\} \tag{4}$$

*where $\mathbf{h}_{i,c}^*$ is the optimized feature representation for the $i$-th sample in category $c$, $n_c$ is the total number of labeled features belonging to class $c$, and $C$ is the total number of categories.*

The proof of Theorem 1 can be found in Appendix A.1. Theorem 1 implies that a primary factor leading to the collapse of features is the use of label encoding, *i.e.*, samples from the same category share the same label encoding. In addition, Theorem 1 is loss-agnostic and solely relies on the mapping property of $f(\cdot)$.

Furthermore, we extend Theorem 1 to the ERM framework. Under this framework, the linear and softmax classifiers are two of the commonly used classification functions. The linear classifier is an injective function, which thus satisfies Theorem 1. The softmax classifier is not an injective function. However, under the constraint that the inputs are non-negative and equimodular, it is injective and satisfies Theorem 1. The above constraint is relatively common, since the non-negative and equimodular conditions can be realized by using the ReLU activation function (Nair & Hinton, 2010) and $l_2$ normalization, respectively. The detailed proof is offered in Appendix A.4.

## 4.2 LABEL-ENCODING RISK MINIMIZATION

Under label insufficient scenarios, we aim to find accurate supervised information that can be utilized to unlabeled samples. Recall that the NCC phenomenon is attributed to the use of label encodings. These label encodings serve as reliable supervised information for learning from labeled samples. Moreover, note that the label encodings remain consistent for both labeled and unlabeled samples under label insufficient scenarios studied in this paper. Consequently, it is reasonable to apply label encodings to guide the learning process of unlabeled samples. Inspired by this, we design a *label-encoding risk*. Specifically, we first calculate the weighted average of prediction probabilities for unlabeled samples in each category, *i.e.*, prediction mean. Based on the properties of prediction means, they can serve as an estimation for label encodings. Then, the label-encoding risk is measured by the divergence between the estimated and ground-truth label encodings for all categories. Next, we detail how to formulate the label-encoding risk.

We first introduce how to calculate the prediction means of unlabeled samples. Let $f(\cdot)$ be a classifier and $g(\cdot)$ be a feature extractor in the form of deep neural networks. Given $n_u$ unlabeled samples $\{\mathbf{x}_i^u\}_{i=1}^{n_u}$, we denote $\widetilde{\mathbf{y}}_{i,:}^u = f(g(\mathbf{x}_i^u)) \in \mathbb{R}^C$ as the prediction probability of $\mathbf{x}_i^u$ provided by $f(\cdot)$ and $g(\cdot)$, where its $c$-th element $\widetilde{y}_{i,c}^u$ denotes the probability (weight) of $\mathbf{x}_i^u$ belonging to category $c$. Accordingly, the prediction mean for unlabeled samples belonging to category $c$ can be calculated by

$$\mathbf{m}_c^u = \frac{1}{\sum_{i=1}^{n_u} \widetilde{y}_{i,c}^u} \Big( \sum_{i=1}^{n_u} \widetilde{y}_{i,c}^u \widetilde{\mathbf{y}}_{i,:}^u \Big). \tag{5}$$

Then, we explain why $\mathbf{m}_c^u$ could be treated as an estimation of label encoding associated with category $c$. To this end, we summarize the properties of $\mathbf{m}_c^u$ in Theorem 2.

**Theorem 2** $\mathbf{m}_c^u$ *satisfies the following properties:*

*(1)* $\mathbf{1}^T \mathbf{m}_c^u = 1$, *where* $\mathbf{1} \in \mathbb{R}^C$ *denotes an all-ones vector.*

*(2)* $0 \leq m_{c,j}^u \leq 1$, $\forall j \in \{1, \ldots, C\}$, *where* $m_{c,j}^u$ *denotes the $j$-th element of* $\mathbf{m}_c^u$.

*(3) If* $\widetilde{\mathbf{y}}_{i,:}^u$ *is the ground-truth label encoding of sample* $\mathbf{x}_i^u$, $\forall i \in \{1, \ldots, n_u\}$, *then* $\mathbf{m}_c^u$ *equals to* $\mathbf{e}_c$, *where* $\mathbf{e}_c$ *denotes the one-hot label encoding associated with category $c$ and is a one-hot vector where its $c$-th element equals 1 while all other elements are 0.*

*(4) If* $\mathbf{m}_c^u$ *is equal to* $\mathbf{e}_c$ *for some* $c \in \{1, \cdots, C\}$, *then for any* $i \in \{1, \cdots, n_u\}$, $\widetilde{\mathbf{y}}_{i,:}^u$ *either equals to* $\mathbf{e}_c$ *or satisfies the condition that* $\widetilde{y}_{i,c}^u = 0$, $0 \leq \widetilde{y}_{i,k}^u \leq 1$, $\forall k \neq c$.

*(5) If* $\mathbf{m}_c^u$ *is equal to* $\mathbf{e}_c$ *for any* $c \in \{1, \cdots, C\}$, *then for any* $i \in \{1, \cdots, n_u\}$, $\widetilde{\mathbf{y}}_{i,:}^u$ *is a one-hot vector with only one entry equal to 1 and other entries being 0.*

The proof of Theorem 2 can be found in Appendix A.2. Based on the property (3) in Theorem 2, we find that when $\widetilde{\mathbf{y}}_{i,:}^u$ approaches the ground-truth label encoding of sample $\mathbf{x}_i^u$, $\mathbf{m}_c^u$ tends to approach label encoding of $\mathbf{e}_c$. Accordingly, $\mathbf{m}_c^u$ could be viewed as an estimation for $\mathbf{e}_c$.

Building upon the above theoretical perspectives, we formulate the label-encoding risk as

$$\mathcal{R}_{ler}(f, g) = \frac{1}{C} \sum_{c=1}^{C} \mathcal{L}(\mathbf{m}_c^u, \mathbf{e}_c), \tag{6}$$

where $\mathcal{L}(\cdot, \cdot)$ denotes the cross-entropy loss. By minimizing the label-encoding risk, we could make estimated label encodings close to the corresponding ground-truth label encodings, which can guide the learning of prediction means for those unlabeled samples. We refer to the minimization process as the LRM, which can be utilized as a plugin for models to handle label insufficient scenarios.

### 4.3 Discussions

**Connection between LRM and NCC.** According to property (5) of Theorem 2, if $\mathbf{m}_c^u$ equals $\mathbf{e}_c$ for any $c \in \{1, \cdots, C\}$ (*i.e.*, $\mathcal{R}_{ler}(f, g) = 0$), the prediction of each unlabeled sample could equal a one-hot vector. This implies that LRM will collapse the prediction for each sample, ultimately resulting in the aforementioned NCC property through Theorem 1.

**Comparison between LRM and EntMin.** On one hand, unlike EntMin which is sample-specific, the LRM is class-specific according to its definition in Eq. (6). Thus, the LRM optimizes all $C$ categories equally, which could help mitigate the dominance of any large category (please refer to Appendix C for detailed analysis and comparison). Accordingly, the LRM considers both the prediction discriminability and diversity. On the other hand, the LRM adopts ground-truth label encodings as the supervision information, in contrast to the soft-labels adopted by EntMin. Moreover, a similar behavior in LRM and EntMin is that they both seek sharp prediction probabilities, which are close to one-hot probabilities. To achieve that, EntMin minimizes the entropy of the prediction probability of each unlabeled sample, while in contrast, according to properties (4) and (5) in Theorem 2, LRM minimizes the label-encoding risk, which is verified in Appendix D.4.

**Connection between LRM and ERM.** As LRM draws inspiration from NCC, a widely observed phenomenon within the ERM framework, we reveal the theoretical relationship between LRM and ERM in the following theorem.

**Theorem 3** *Under the setting of supervised learning, we assume that the corresponding dataset is balanced among classes. If the loss function is convex or strongly convex, then the label-encoding risk is upper-bounded by the empirical risk.*

The proof of Theorem 3 is offered in Appendix A.3. Theorem 3 implies that under the setting of class-balanced supervised learning, there is a relationship between the empirical and label-encoding risks. Moreover, experimental results in Section 5.2 verify Theorem 3 empirically.

### 4.4 APPLICATION TO LABEL INSUFFICIENT SCENARIOS

**Application to SSL**. Under the SSL setting, we have $n_l$ labeled sample $\{(\mathbf{x}_i^l, \mathbf{y}_i^l)\}_{i=1}^{n_l}$, where $\mathbf{y}_i^l$ is the one-hot label for $\mathbf{x}_i^l$. Also, we have $n_u$ unlabeled samples $\{\mathbf{x}_i^u\}_{i=1}^{n_u}$. Here, we have $n_l \ll n_u$. The goal is to pick an excellent model for predicting $\{\mathbf{x}_i^u\}_{i=1}^{n_u}$. To this end, we apply the ERM and LRM to labeled and unlabeled samples, respectively. In addition, following the setting of supervised pre-training in (Chen et al., 2022), we augment the labeled and unlabeled samples by leveraging a weak augmentation function, *i.e.*, $\psi(\cdot)$, and a strong augmentation, *i.e.*, $\Psi(\cdot)$, respectively. Thus, the overall objective function can be formulated as

$$\min_{f,g} \frac{1}{n_l} \sum_{i=1}^{n_l} \mathcal{L}\Big(f\big(g(\psi(\mathbf{x}_i^l))\big), \mathbf{y}_i^l\Big) + \frac{\mu}{n_l} \sum_{i=1}^{n_l} \mathcal{L}\Big(f\big(g(\Psi(\mathbf{x}_i^l))\big), \mathbf{y}_i^l\Big) + \frac{\lambda}{C} \sum_{c=1}^{C} \Big[\mathcal{L}(\mathbf{w}_c^u, \mathbf{e}_c) + \mu\mathcal{L}(\mathbf{s}_c^u, \mathbf{e}_c)\Big], \quad (7)$$

where $g(\cdot)$, $f(\cdot)$, and $\mathcal{L}(\cdot, \cdot)$ are the feature extractor, softmax classifier, and cross-entropy loss, respectively, $\lambda$ and $\mu$ are two trade-off hyperparameters, and the calculation of $\mathbf{w}_c^u$ and $\mathbf{s}_c^u$ is similar to Eq. (5) by replacing $\mathbf{x}_i^u$ in $\mathbf{m}_c^u$ with $\psi(\mathbf{x}_i^u)$ and $\Psi(\mathbf{x}_i^u)$, respectively.

**Application to UDA**. For the UDA task, we are offered $n_s$ labeled source samples $\{(\mathbf{x}_i^l, \mathbf{y}_i^l)\}_{i=1}^{n_s}$ and $n_t$ unlabeled target samples $\{\mathbf{x}_i^u\}_{i=1}^{n_t}$. The goal is to find a high-quality model for categorizing $\{\mathbf{x}_i^u\}_{i=1}^{n_t}$. To achieve this, we apply the ERM and LRM to labeled source and unlabeled target samples, respectively. Accordingly, we can formulate the objective function as

$$\min_{f,g} \frac{1}{n_s} \sum_{i=1}^{n_s} \mathcal{L}\Big(f(g(\mathbf{x}_i^l)), \mathbf{y}_i^l\Big) + \frac{\lambda}{C} \sum_{c=1}^{C} \mathcal{L}(\mathbf{m}_c^u, \mathbf{e}_c), \quad (8)$$

where $g(\cdot)$ is the feature extractor, $f(\cdot)$ is the classifier, and $\mathbf{m}_c^u$ can be obtained in Eq. (5).

**Application to SHDA**. In the SHDA problem, we are given $n_l$ labeled target samples $\{(\mathbf{x}_i^l, \mathbf{y}_i^l)\}_{i=1}^{n_l}$, $n_s$ labeled source samples $\{(\mathbf{x}_i^s, \mathbf{y}_i^s)\}_{i=1}^{n_s}$ and $n_t$ unlebeled target samples $\{\mathbf{x}_i^u\}_{i=1}^{n_t}$. Here, we have $n_s \gg n_l$ and $n_t \gg n_l$. The goal is to learn a model for classifying $\{\mathbf{x}_i^u\}_{i=1}^{n_t}$. As the heterogeneity of the features across domains, we let $g_s(\cdot)$ and $g_t(\cdot)$ represent the feature extractors in the source and target domains, respectively, and let $f(\cdot)$ be the classifier. For the above purpose, we apply the ERM to labeled source and target samples, and utilize the LRM on unlabeled target samples. In addition, following the setting in (Yao et al., 2019), an additional regulation term is put to use to prevent overfitting. As a result, we can derive the following objective function as

$$\min_{f,g_s,g_t} \frac{1}{n_s} \sum_{i=1}^{n_s} \mathcal{L}\Big(f(g_s(\mathbf{x}_i^s)), \mathbf{y}_i^s\Big) + \frac{1}{n_l} \sum_{i=1}^{n_l} \mathcal{L}\Big(f(g_t(\mathbf{x}_i^l)), \mathbf{y}_i^l\Big) + \frac{\lambda}{C} \sum_{c=1}^{C} \mathcal{L}(\dot{\mathbf{m}}_c^u, \mathbf{e}_c) + \tau(\|f\|^2 + \|g_s\|^2 + \|g_t\|^2),$$
$$(9)$$

where the calculation of $\dot{\mathbf{m}}_c^u$ is similar to Eq. (5) by replacing $g(\cdot)$ in $\mathbf{m}_c^u$ with $g_t(\cdot)$, and $\lambda$, $\tau$ act as two trade-off hyperparameters.

## 5 EXPERIMENTS

We evaluate the LRM on three typical label insufficient scenarios, including SSL, UDA, and SHDA.

### 5.1 EVALUATION

**Evaluation on SSL Tasks**. We evaluate the LRM on three SSL benchmark datasets, including CIFAR-10 (Krizhevsky et al., 2009), CIFAR-100 (Krizhevsky et al., 2009), and DTD (Cimpoi et al., 2014). The CIFAR-10 and CIFAR-100 datasets consist of 60,000 images with a resolution of $32 \times 32$

Table 1: Accuracy (%) comparison on the CIFAR-10, CIFAR-100, and DTD datasets for SSL. The best performance of each task is marked in bold and the best performance in each comparison group is underlined.

| Dataset | CIFAR-10 | | | CIFAR-100 | | | DTD | | |
|---|---|---|---|---|---|---|---|---|---|
| # Label per category | 1 | | 4 | 1 | | 4 | 1 | | 4 |
| | Top-1 | Top-5 | Top-1 | Top-1 | Top-5 | Top-1 | Top-1 | Top-5 | Top-1 |
| ERM (Vapnik, 1999) | 32.24 | 78.16 | 57.04 | 23.58 | 47.51 | 47.18 | 31.22 | 58.99 | 50.66 |
| ERM+EntMin (Grandvalet & Bengio, 2004) | 28.17 | 71.05 | 59.62 | 15.32 | 43.95 | 45.40 | 21.55 | 51.65 | 50.96 |
| ERM+BNM (Cui et al., 2020) | 27.02 | 70.37 | 52.46 | 21.79 | 47.72 | 58.90 | 28.55 | 54.61 | 48.26 |
| ERM+LRM | 36.00 | 80.92 | 76.94 | 31.77 | 61.39 | 59.52 | 36.65 | 65.78 | 54.34 |
| FlexMatch (Zhang et al., 2021a) | 40.86 | 84.75 | 86.66 | 16.49 | 42.40 | 65.11 | 33.39 | 58.48 | 54.96 |
| FlexMatch+EntMin | 43.79 | 87.69 | 86.56 | 13.00 | 42.83 | 67.32 | 32.20 | 58.49 | 54.91 |
| FlexMatch+BNM | 41.95 | 78.73 | 86.57 | 15.04 | 43.54 | 64.46 | 31.31 | 57.31 | 55.04 |
| FlexMatch+LRM | 56.86 | 93.01 | 87.75 | 20.28 | 42.46 | 68.04 | 34.22 | 59.56 | 55.51 |
| DST (Chen et al., 2022) | 51.11 | 91.76 | 88.05 | 32.92 | 64.65 | 66.80 | 34.88 | 61.99 | 56.40 |
| DST+EntMin | 45.46 | 92.41 | 87.85 | 25.48 | 60.92 | 66.79 | 32.32 | 62.27 | 56.13 |
| DST+BNM | 55.03 | 91.75 | 88.49 | 32.15 | 65.16 | 67.27 | 36.08 | 64.06 | 56.51 |
| DST+LRM | **67.91** | **95.90** | **89.48** | **38.81** | **69.51** | **70.41** | **37.63** | **65.59** | **57.04** |

pixels, categorized into 10 and 100 categories, respectively. The DTD dataset contains 5,640 textural images in 47 categories. We conduct experiments on the SSL tasks with limited labeled samples, *i.e.*, one and four labeled samples per category. In the comparison experiments, we combine the LRM with ERM, and state-of-the-art SSL approaches such as FlexMatch (Zhang et al., 2021a) and DST (Chen et al., 2022) to tackle all the above tasks. Also, we realize the EntMin and BNM in the same way as the LRM, and report the average classification accuracy of each method in three randomized trials. Due to page limit, more experimental details can be found in the Appendix B.1.

According to the results shown in Table 1, we can see that LRM yields significant performance improvements under all the settings. Specifically, for the SSL tasks with four labeled samples per category, the LRM achieves accuracy improvements of 19.90%, 12.34%, and 3.68% over the ERM method on the CIFAR-10, CIFAR-100, and DTD datasets, respectively, and it also performs better than EntMin and BNM. Moreover, when combined with state-of-the-art SSL methods, the LRM could further improve the performance. For instance, LRM brings performance improvements of 2.93% and 3.61% over FlexMatch and DST on the CIFAR-100 dataset, respectively. Those results highlight the potential benefits of combining LRM with existing SSL approaches. For the more challenging case with one labeled sample per category, both the EntMin and BNM exhibit varying degrees of performance degradation when compared with ERM. On the contrary, the LRM shows consistent performance improvements on three benchmark datasets, which demonstrates its effectiveness in scenarios with extremely limited labeled samples. Moreover, the LRM further enhances the performance of FlexMatch and DST, and surpasses other methods with a large margin in terms of top-1 accuracy. Specifically, with only one labeled sample per category, LRM achieves a top-1 performance improvement of 16.8% over the DST method on the CIFAR-10 dataset, demonstrating its superiority for better utilization of unlabeled samples. In summary, LRM has shown promising results in improving the performance of existing SSL methods and outperforms EntMin and BNM.

**Evaluation on UDA Tasks**. We evaluate the LRM on two UDA benchmark datasets, *i.e.*, Office-31 (Saenko et al., 2010) and Office-Home (Venkateswara et al., 2017). The Office-31 dataset contains three domains: Amazon (A), DSLR (D), and Webcam (W), with 4,110 images in 31 categories. We evaluate six transfer tasks built on the above domains. The Office-Home dataset contains about 15,500 images from 65 categories within four domains: Art (Ar), Clipart (Cl), Product (Pr), and Real-World (Rw). By utilizing these four domains, we construct 12 transfer tasks for evaluation. We combine the LRM with ERM, and state-of-the-art UDA methods such as CDAN and SDAT for dealing with all the above tasks. Additionally, we implement the EntMin and BNM in the same fashion as the LRM. For each method, we run three random experiments and list the average classification accuracy. Due to page limit, more experimental details are offered in the Appendix B.2.

The results on the Office-31 dataset are shown in Table 2. As can be seen, the LRM brings a substantial performance improvement of 7.77% over ERM, and surpasses EntMin and BNM by a large margin of 4.11% and 1.17%, respectively. Remarkably, ERM + LRM even performs better than well-established UDA methods such as DANN, AFN, and CDAN. Note that the LRM does not explicitly match the distributions across domains. Instead, the LRM aligns the prediction means for target samples and their corresponding label encodings, while the ERM reduces the divergence

Table 2: Accuracy (%) comparison on the Office-31 dataset for UDA. The best performance of each task is marked in bold and the best performance in each comparison group is underlined.

| Method | A→D | A→W | D→W | W→D | D→A | W→A | Average |
|---|---|---|---|---|---|---|---|
| ERM (Vapnik, 1999) | 81.15 | 77.00 | 96.60 | 99.00 | 63.98 | 64.01 | 80.29 |
| DANN (Ganin et al., 2016) | 83.62 | 89.30 | 97.81 | **100.00** | 72.01 | 74.11 | 86.14 |
| AFN (Xu et al., 2019) | 95.29 | 91.18 | 98.73 | **100.00** | 72.13 | 70.60 | 87.99 |
| ERM+EntMin (Grandvalet & Bengio, 2004) | 87.42 | 88.01 | 98.49 | 99.93 | 68.04 | 61.83 | 83.95 |
| ERM+BNM (Cui et al., 2020) | 89.36 | 91.36 | 98.62 | 99.93 | 70.76 | 71.29 | 86.89 |
| ERM+LRM | 92.26 | 92.83 | 98.74 | 99.95 | 72.21 | 72.35 | 88.06 |
| CDAN (Long et al., 2018) | 92.77 | 92.37 | 98.79 | **100.00** | 72.46 | 70.31 | 87.78 |
| CDAN+EntMin | 92.64 | 91.49 | 98.87 | **100.00** | 71.87 | 71.80 | 87.78 |
| CDAN+BNM | 92.17 | 92.87 | 99.20 | **100.00** | 73.53 | 73.15 | 88.49 |
| CDAN+LRM | 93.88 | 92.92 | 99.12 | **100.00** | 74.35 | 74.63 | 89.15 |
| SDAT (Rangwani et al., 2022) | 94.99 | 89.77 | **99.04** | **100.00** | 77.04 | 72.73 | 88.93 |
| SDAT+EntMin | 95.58 | 93.04 | 98.74 | **100.00** | 77.50 | 72.41 | 89.54 |
| SDAT+BNM | 95.58 | 92.91 | 98.66 | **100.00** | 77.61 | 74.97 | 89.96 |
| SDAT+LRM | **96.65** | **93.31** | 98.95 | **100.00** | **77.98** | **75.18** | **90.34** |

between the predicted labels for the source samples and their ground-truth labels. Thus, ERM + LRM implicitly reduces the distributional divergence across domains in a classification manner, leading to better transfer performance. This is one important reason for the above results. Moreover, as can be seen, the performance of CDAN + LRM outperforms that of CDAN + EntMin and CDAN + BNM by 1.37% and 0.66%, respectively. Similar observations can be found in SDAT, highlighting the potential of the LRM. Due to page limit, experiments on the Office-Home datasets are put in Appendix D.1. In a nutshell, the LRM shows transferability by combining existing UDA (or ERM alone) methods.

**Evaluation on SHDA Tasks**. We evaluate the LRM on the SHDA tasks involving text-to-text and text-to-image scenarios. For the former, we adopt the Multilingual Reuters Collection dataset (Amini et al., 2009), which consists of about 11,000 articles from six categories written in English (E), French (F), German (G), Italian (I), and Spanish (S). Following (Duan et al., 2012; Hsieh et al., 2016; Li et al., 2013; Yao et al., 2019), we treat S as the target domain, and the remaining as the source ones. For the latter, we follow (Chen et al., 2016; Yao et al., 2019) and use the NUS-WIDE (N) (Chua et al., 2009) and ImageNet (I) (Deng et al., 2009) datasets as the source and target domains, respectively. Eight shared categories from both domains are chosen. Also, for the source domain, we randomly select 100 texts per category as the labeled samples. As for the target domain, we randomly single out three images from each category as the labeled samples, and the remaining images are considered as the unlabeled samples. We combine the LRM with ERM, and state-of-the-art SHDA approach *i.e.*, KPG[2] (Gu et al., 2022), to handle all the above tasks. For a fair comparison, we implement the EntMin and BNM in the same manner as the LRM, and present the average classification accuracy of each method in five random experiments. More experimental details are given in Appendix B.3.

Table 3 lists the results on SHDA tasks, where S5 and S10 denote the number of labeled target samples per category are five and ten, respectively. We can observe that compared to EntMin and BNM, both ERM and KPG demonstrate the most significant performance improvements when combined with LRM. In particular, the classification accuracy of ERM + LRM on the N→I is 79.83%, which exceeds ERM, ERM+EntMin, and ERM+BNM by 10.86%, 10.26%, and 1.5%, respectively. Overall, these results verify that even in heterogeneous scenarios, the LRM is still effective and superior.

## 5.2 ANALYSIS

**Effectiveness and Convergence**. We evaluate the effectiveness and convergence of the LRM on the SHDA task of E→S5. Specifically, we plot the loss values of ERM and LRM *w.r.t.* the ERM and ERM + LRM in Figure 2(a), respectively. We also show the accuracy curves of both methods in Figure 2(a). We have several observations. (1) The loss values of ERM in both methods have experienced notable reductions, as they explicitly minimize the ERM loss in their objective functions. Also, the loss value of ERM in ERM + LRM is slightly lower than that of ERM in ERM alone, which

---

[2]Since KPG uses an SVM classifier, for convenience, we first extract the learned features of KPG and then use a softmax classifier for categorizing.

Table 3: Accuracy (%) comparison on the text-to-text and text-to-image datasets for SHDA. The best performance of each task is marked in bold and the best performance in each comparison group is underlined.

| Method | E→S5 | F→S5 | G→S5 | I→S5 | Average | E→S10 | F→S10 | G→S10 | I→S10 | Average | N→I |
|---|---|---|---|---|---|---|---|---|---|---|---|
| ERM | 61.62 | 61.75 | 60.91 | 62.11 | 61.60 | 68.53 | 68.47 | 68.36 | 68.96 | 68.58 | 68.97 |
| ERM+EntMin | 62.31 | 61.99 | 61.13 | 61.72 | 61.78 | 70.80 | 70.32 | 70.79 | 71.68 | 70.90 | 69.57 |
| ERM+BNM | 69.73 | 68.77 | 70.23 | 69.93 | 69.66 | 73.49 | 73.41 | 73.49 | 73.79 | 73.55 | 78.33 |
| ERM+LRM | **70.14** | **68.80** | **70.82** | **70.08** | **69.96** | **74.63** | 74.43 | **74.43** | 73.97 | **74.36** | **79.83** |
| KPG+ERM | 62.23 | 61.32 | 61.53 | 61.29 | 61.59 | 67.11 | 64.02 | 64.69 | 63.99 | 64.95 | 61.68 |
| KPG+BNM | 65.55 | 65.62 | 65.11 | 65.15 | 65.36 | 73.09 | 73.78 | 73.23 | 73.47 | 73.39 | 74.52 |
| KPG+LRM | 68.36 | 68.39 | 68.20 | 67.99 | 68.24 | 73.80 | 74.47 | 73.79 | 74.11 | 74.04 | 77.25 |

indicates that the LRM may assist in further reducing the loss value of ERM. (2) When ERM is combined with LRM, the accuracy curve improves by a large margin, which implies that LRM can effectively improve the performance. (3) The loss value of LRM in ERM + LRM is significantly lower than that of LRM in ERM, which is reasonable since ERM does not take LRM into account. Meanwhile, it is also one important reason for the observation of (2). (4) The loss value of LRM in ERM + LRM first decreases gradually and then hardly changes as the number of iterations grows. Also, the accuracy first improves monotonically and then becomes stable as more iterations are conducted. Both imply the convergence of the LRM.

**Connection between the LRM and ERM**. In the SHDA task of N→I, as aforementioned, there are 100 labeled source samples and three labeled target samples per category. Accordingly, under the setting where only labeled samples from both domains are available for training, which can be regarded as a class-balanced supervised learning task. Hence, we empirically verify the connection between LRM and ERM under this setting. We calculate the loss values of ERM and LRM on all labeled samples, respectively. Please refer to Appendix A.3 for the calcu-

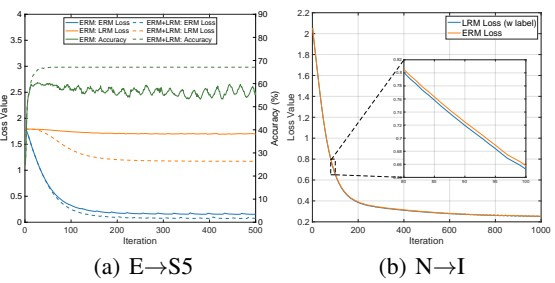

(a) E→S5      (b) N→I

Figure 2: Comparison between LRM and ERM.

lation of the loss value of LRM on all labeled samples. Figure 2(b) highlights that the loss value of ERM consistently surpasses that of LRM during the training. This confirms that ERM is an upper bound for the LRM under the utilization of the convex cross-entropy loss, thereby providing experimental evidence for Theorem 3.

**Parameter Sensitivity and Feature Visualization**. We conclude the analysis of parameter sensitivity and feature visualization in Appendix D. These results indicate that the LRM is insensitive to different tasks, and can produce effective transfer in combination with the ERM.

# 6    CONCLUSION

In this paper, we reveal the key rationale behind the NCC, and propose the LRM based on this inspiration for handling label insufficient scenarios. The LRM can be seen as an extension of the ERM to unlabeled samples. It adopts label encodings as supervision information, and the prediction means are designed to estimate the label encodings for unlabeled samples, which ensures both the prediction discriminability and diversity. Also, under the setting of class-balanced supervised learning, we theoretically analyze the relationship between the LRM and ERM. Experiments on three typical scenarios with insufficient labeled samples validate the effectiveness of the LRM. As a future direction, we intend to investigate the relationship between the LRM and ERM, in the context of class-imbalanced supervised learning. Moreover, we believe that the LRM has the potential to serve as an elegant and effective alternative to EntMin, thereby opening a new door for tackling unlabeled samples. Hence, applying the LRM to other label insufficient scenarios is also our interest.

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

# A  THEORETICAL ANALYSES

## A.1  PROOF FOR THEOREM 1

Since the learned features belonging to the same category collapse to the class mean, when $\mathcal{L}(f(\mathbf{H}), \mathbf{Y})$ *w.r.t.* $\mathbf{H}$ and $f$ is minimized to 0, we have

$$f^*(\mathbf{h}_{1,c}^*) = \cdots = f^*(\mathbf{h}_{n_c,c}^*) = \mathbf{e}_c, \forall c \in \{1, \ldots, C\}, \tag{10}$$

Since $f(\cdot)$ is a injective function, we obtain

$$\mathbf{h}_{1,c}^* = \cdots = \mathbf{h}_{n_c,c}^*. \tag{11}$$

Hence, we have

$$\frac{1}{n_c} \sum_{i=1}^{n_c} \mathbf{h}_{i,c}^* = \frac{1}{n_c} \sum_{i=1}^{n_c} \mathbf{h}_{1,c}^* = \mathbf{h}_{1,c}^* = \cdots = \mathbf{h}_{n_c,c}^*, \forall c \in \{1, \ldots, C\}. \tag{12}$$

## A.2  PROOF FOR THEOREM 2

(1) The sum of all elements in $\mathbf{m}_c^u$, *i.e.*, $\mathbf{1}^T \mathbf{m}_c^u$, can be calculated as follows:

$$\mathbf{1}^T \mathbf{m}_c^u = \frac{\mathbf{1}^T \sum_{i=1}^{n_u} (\widetilde{y}_{i,c}^u \widetilde{\mathbf{y}}_{i,:}^u)}{\sum_{i=1}^{n_u} \widetilde{y}_{i,c}^u} = \frac{\sum_{i=1}^{n_u} (\widetilde{y}_{i,c}^u (\mathbf{1}^T \widetilde{\mathbf{y}}_{i,:}^u))}{\sum_{i=1}^{n_u} \widetilde{y}_{i,c}^u} = \frac{\sum_{i=1}^{n_u} \widetilde{y}_{i,c}^u}{\sum_{i=1}^{n_u} \widetilde{y}_{i,c}^u} = 1. \tag{13}$$

(2) Since $0 \leq \widetilde{y}_{i,j}^u \leq 1, \forall j \in \{1, \ldots, C\}, \forall i \in \{1, \ldots, n_u\}$, based on Eq. (5) and the Property (1) in Theorem 2, we have

$$0 \leq m_{c,j}^u \leq 1, \forall j \in \{1, \ldots, C\}. \tag{14}$$

(3) Since $\widetilde{\mathbf{y}}_{i,:}^u$ is the ground-truth label encoding of sample $\mathbf{x}_i^u$, $\widetilde{y}_{i,c}^u = 1$ if $\mathbf{x}_i^u$ belongs to category $c$, else $\widetilde{y}_{i,c}^u = 0$. Accordingly, we can calculate the $c$-th element of $\mathbf{m}_c^u$ as follows:

$$m_{c,c}^u = \frac{\sum_{i=1}^{n_u} \widetilde{y}_{i,c}^u \widetilde{y}_{i,c}^u}{\sum_{i=1}^{n_u} \widetilde{y}_{i,c}^u} = \frac{n_u^c}{n_u^c} = 1. \tag{15}$$

Similarly, the $k$-th ($\forall k \neq c$) element of $\mathbf{m}_c^u$ can be calculated as follows:

$$m_{c,k}^u = \frac{\sum_{i=1}^{n_u} \widetilde{y}_{i,c}^u \widetilde{y}_{i,k}^u}{\sum_{i=1}^{n_u} \widetilde{y}_{i,c}^u} = \frac{0}{n_u^c} = 0, \forall k \neq c. \tag{16}$$

Hence, $\mathbf{m}_c^u$ equals to $\mathbf{e}_c$.

(4) Since $\mathbf{m}_c^u = \mathbf{e}_c$ for some $c \in \{1, \cdots, C\}$, we have $m_{c,c}^u = 1$ and $m_{c,k}^u = 0, \forall k \neq c$. According to the constraint of $m_{c,c}^u = 1$, we obtain

$$\begin{aligned} m_{c,c}^u = \frac{\sum_{i=1}^{n_u} \widetilde{y}_{i,c}^u \widetilde{y}_{i,c}^u}{\sum_{i=1}^{n_u} \widetilde{y}_{i,c}^u} = 1 &\Rightarrow \sum_{i=1}^{n_u} (\widetilde{y}_{i,c}^u)^2 - \sum_{i=1}^{n_u} \widetilde{y}_{i,c}^u = 0 \\ &\Rightarrow (\widetilde{\mathbf{y}}_{:,c}^u)^T \widetilde{\mathbf{y}}_{:,c}^u - \mathbf{1}^T \widetilde{\mathbf{y}}_{:,c}^u = 0 \\ &\Rightarrow (\widetilde{\mathbf{y}}_{:,c}^u - \mathbf{1})^T (\widetilde{\mathbf{y}}_{:,c}^u - \mathbf{0}) = 0, \end{aligned} \tag{17}$$

where $\widetilde{\mathbf{y}}_{:,c}^u$ denotes the prediction probabilities of $\{\mathbf{x}_i^u\}_{i=1}^{n_u}$ belonging to category $c$, and $\mathbf{1} \in \mathbb{R}^{n_u}$ denotes an all-ones vector, and $\mathbf{0} \in \mathbb{R}^{n_u}$ denotes an all-zero vector. Since $0 \leq \widetilde{y}_{i,c}^u \leq 1, \forall i \in \{1, \cdots, n_u\}$, we can get

$$\widetilde{y}_{i,c}^u = 1 \quad \text{or} \quad \widetilde{y}_{i,c}^u = 0, \forall i \in \{1, \cdots, n_u\}. \tag{18}$$

Based on the constraint of $m_{c,k}^u = 0, \forall k \neq c$, we have

$$m_{c,k}^u = \frac{\sum_{i=1}^{n_u} \widetilde{y}_{i,c}^u \widetilde{y}_{i,k}^u}{\sum_{i=1}^{n_u} \widetilde{y}_{i,c}^u} = 0 \Rightarrow \widetilde{y}_{i,c}^u \widetilde{y}_{i,k}^u = 0 \Rightarrow \widetilde{y}_{i,c}^u = 0 \quad \text{or} \quad \widetilde{y}_{i,k}^u = 0, \forall k \neq c. \tag{19}$$

According to Eq. (18) and Eq. (19), for some $c \in \{1, \cdots, C\}$, and any $i \in \{1, \cdots, n_u\}$, we obtain

$$\widetilde{y}_{i,c}^u = 1, \widetilde{y}_{i,k}^u = 0, \forall k \neq c \Rightarrow \widetilde{\mathbf{y}}_{i,:}^u = \mathbf{e}_c \quad \textbf{or} \quad \widetilde{y}_{i,c}^u = 0, 0 \leq \widetilde{y}_{i,k}^u \leq 1, \forall k \neq c. \tag{20}$$

(5) Since $\mathbf{m}_c^u$ is equal to $\mathbf{e}_c$ for any $c \in \{1, \cdots, C\}$, based on the Property (4), for any $i \in \{1, \cdots, n_u\}$, we have

$$\widetilde{y}_{i,c}^u = 0 \quad \textbf{or} \quad \widetilde{y}_{i,c}^u = 1, \forall c \in \{1, \cdots, C\}. \tag{21}$$

Hence, $\widetilde{\mathbf{y}}_{i,:}^u$ is a one-hot vector for any $i \in \{1, \cdots, n_u\}$.

### A.3 PROOF FOR THEOREM 3

Given $n_l$ labeled sample $\{(\mathbf{x}_i^l, \mathbf{y}_i^l)\}_{i=1}^{n_l}$, where $\mathbf{y}_i^l$ is the one-hot label for $\mathbf{x}_i^l$. Since the dataset is balanced across categories, we let $n$ be the number of labeled samples in each class, we have $n_l = nC$. In addition, we denote by $f(\cdot) : \mathbb{R}^d \rightarrow \mathbb{R}^C$ a classification function, and by $\mathcal{L}(\cdot, \cdot)$: $\mathbb{R}^C \times \mathbb{R}^C \rightarrow [0, +\infty)$ a loss function satisfying that $\mathcal{L}(\mathbf{y}, \mathbf{z}) = 0$ indicates $y_i = z_i, \forall i \in \{1, \ldots, C\}$.

The empirical risk is given by

$$\mathcal{R}_{emp}(f) = \frac{1}{n_l} \sum_{i=1}^{n_l} \mathcal{L}(\widetilde{\mathbf{y}}_i^l, \mathbf{y}_i^l) = \frac{1}{nC} \sum_{c=1}^{C} \sum_{j=1}^{n} \mathcal{L}(\widetilde{\mathbf{y}}_{j,c}^l, \mathbf{e}_c), \tag{22}$$

where $\mathbf{e}_c$ denotes the label encoding associated with category $c$, $\widetilde{\mathbf{y}}_i^l = f(\mathbf{x}_i^l) \in \mathbb{R}^C$ is the prediction probability of $\mathbf{x}_i^l$, $\widetilde{\mathbf{y}}_{j,c}^l = f(\mathbf{x}_{j,c}^l)$ is the prediction probability of $\mathbf{x}_{j,c}^l$. Here, $\mathbf{x}_{j,c}^l$ is $j$-th labeled sample belonging to category $c$.

The label-encoding risk is given by

$$\mathcal{R}_{ler}(f) = \frac{1}{C} \sum_{c=1}^{C} \mathcal{L}(\mathbf{m}_c^l, \mathbf{e}_c), \tag{23}$$

where $\mathbf{m}_c^l$ is the prediction mean of labeled samples belonging to category $c$, which can be calculated by

$$\mathbf{m}_c^l = \frac{1}{\sum_{i=1}^{n_l} \mathbf{y}_{i,c}^l} \left( \sum_{i=1}^{n_l} \mathbf{y}_{i,c}^l \widetilde{\mathbf{y}}_{i,:}^l \right) = \frac{1}{n} \sum_{j=1}^{n} \widetilde{\mathbf{y}}_{j,c}^l. \tag{24}$$

Hence, when the loss function $\mathcal{L}(\cdot, \cdot)$ is convex, we have

$$
\begin{aligned}
\mathcal{R}_{emp}(f) &= \frac{1}{nC} \sum_{c=1}^{C} \sum_{j=1}^{n} \mathcal{L}(\widetilde{\mathbf{y}}_{j,c}^l, \mathbf{e}_c) \\
&= \frac{1}{C} \sum_{c=1}^{C} \left[ \frac{1}{n} \sum_{j=1}^{n} \mathcal{L}(\widetilde{\mathbf{y}}_{j,c}^l, \mathbf{e}_c) \right] \\
&\geq \frac{1}{C} \sum_{c=1}^{C} \left[ \mathcal{L}(\frac{1}{n} \sum_{j=1}^{n} \widetilde{\mathbf{y}}_{j,c}^l, \mathbf{e}_c) \right] \\
&= \frac{1}{C} \sum_{c=1}^{C} \mathcal{L}(\mathbf{m}_c^l, \mathbf{e}_c) \\
&= \mathcal{R}_{ler}(f).
\end{aligned}
\tag{25}
$$

The inequality is followed from the convexity of $\mathcal{L}(\cdot, \cdot)$. When $\mathcal{L}(\cdot, \cdot)$ is strongly convex, we obtain $\mathcal{R}_{emp}(f) > \mathcal{R}_{ler}(f)$. Therefore, it proves that when the dataset is balanced among classes, the label-encoding risk is upper-bounded by the empirical risk under the condition that $\mathcal{L}(\cdot, \cdot)$ is convex or strongly convex.

## A.4 THEORETICAL ANALYSIS FOR THE MAPPING PROPERTY OF THE SOFTMAX FUNCTION

**Theorem 4** *Let $\varphi(\cdot) : \mathbb{R}^d \to \mathbb{R}^d$ be a softmax function. If the inputs of $\varphi(\cdot)$ are non-negative and equimodular, then $\varphi(\cdot)$ is injective.*

First, we prove the following Lemma 1.

**Lemma 1** *Let $\varphi(\cdot) : \mathbb{R}^d \to \mathbb{R}^d$ be a softmax function, for any given $\mathbf{h}, \mathbf{k} \in \mathbb{R}^d$, if $\varphi(\mathbf{h}) = \varphi(\mathbf{k})$, then $\exists c \in \mathbb{R}$ such that $\mathbf{k} = \mathbf{h} + c\mathbf{1}$, where $\mathbf{1} \in \mathbb{R}^d$ denotes an all-ones vector.*

*Proof.* Let $h_i$ and $k_i$ denote the $i$-th element in $\mathbf{h}$ and $\mathbf{k}$, respectively, $\forall i \in \{1, \ldots, d\}$, if $\varphi(\mathbf{h}) = \varphi(\mathbf{k})$, we have

$$\varphi(h_i) = \frac{e^{h_i}}{\sum_{j=1}^{d} e^{h_j}} = \frac{e^{k_i}}{\sum_{j=1}^{d} e^{k_j}} = \varphi(k_i). \tag{26}$$

Then, we obtain

$$1 = \frac{\varphi(k_i)}{\varphi(h_i)} = \frac{\frac{e^{k_i}}{\sum_{j=1}^{d} e^{k_j}}}{\frac{e^{h_i}}{\sum_{j=1}^{d} e^{h_j}}} = \frac{e^{k_i}}{\sum_{j=1}^{d} e^{k_j}} \cdot \frac{\sum_{j=1}^{d} e^{h_j}}{e^{h_i}} = e^{k_i - h_i} \frac{\sum_{j=1}^{d} e^{h_j}}{\sum_{j=1}^{d} e^{k_j}}, \tag{27}$$

where $\sum_{j=1}^{d} e^{h_j}$ and $\sum_{j=1}^{d} e^{k_j}$ are fixed when $\mathbf{h}$ and $\mathbf{k}$ are given, respectively. Hence,

$$e^{k_i - h_i} = e^{k_j - h_j}, \forall i, j \in \{1, \ldots, d\}. \tag{28}$$

Accordingly, we have

$$\begin{aligned}
&k_i - h_i = k_j - h_j = c, \forall i, j \in \{1, \ldots, d\}, \\
\Rightarrow\ &h_i + c = k_i, h_j + c = k_j, \forall i, j \in \{1, \ldots, d\}, \\
\Rightarrow\ &\mathbf{h} + c\mathbf{1} = \mathbf{k},
\end{aligned} \tag{29}$$

where $c \in \mathbb{R}$ is a constant. □

Based on Lemma 1, in the following we can further prove Theorem 4. Since the inputs are non-negative, we have

$$k_i = h_i + c \geq 0 \Rightarrow c \geq -h_i \Rightarrow \sum_{i}^{d} c \geq -\sum_{i=1}^{d} h_i \Rightarrow c \geq -\frac{\sum_{i=1}^{d} h_i}{d}, \forall i \in \{1, \ldots, d\}. \tag{30}$$

Also, as the inputs are equimodular, we obtain

$$\begin{aligned}
\|\mathbf{h}\|^2 = \|\mathbf{k}\|^2 = \|\mathbf{h} + c\mathbf{1}\|^2 \Rightarrow\ &\mathbf{h}^{\mathrm{T}}\mathbf{h} = (\mathbf{h} + c\mathbf{1})^{\mathrm{T}}(\mathbf{h} + c\mathbf{1}) \\
\Rightarrow\ &\mathbf{h}^{\mathrm{T}}\mathbf{h} = \mathbf{h}^{\mathrm{T}}\mathbf{h} + c\mathbf{h}^{\mathrm{T}}\mathbf{1} + c\mathbf{1}^{\mathrm{T}}\mathbf{h} + c^2\mathbf{1}^{\mathrm{T}}\mathbf{1} \\
\Rightarrow\ &c(\mathbf{h}^{\mathrm{T}}\mathbf{1} + \mathbf{1}^{\mathrm{T}}\mathbf{h} + c\mathbf{1}^{\mathrm{T}}\mathbf{1}) = 0 \\
\Rightarrow\ &c\Big(2\sum_{i=1}^{d} h_i + cd\Big) = 0.
\end{aligned} \tag{31}$$

Hence we can get

$$c = 0 \quad \textbf{or} \quad c = -\frac{2\sum_{i=1}^{d} h_i}{d}. \tag{32}$$

But the second one is impossible except for $\mathbf{h} = \mathbf{k} = \mathbf{0}$. According to Eq. (30), we have

$$c = -\frac{2\sum_{i=1}^{d} h_i}{d} \geq -\frac{\sum_{i=1}^{d} h_i}{d}. \tag{33}$$

If $\sum_{i=1}^{d} h_i \neq 0$, together with $h_i \geq 0$, $\forall i \in \{1, \ldots, d\}$, the inequality does not hold. Hence $\sum_{i=1}^{d} h_i = 0$, and it implies $\mathbf{h} = \mathbf{0}$ since $h_i$ are non-negative. In this case, $\|\mathbf{k}\|^2 = \|\mathbf{h}\|^2 = \|\mathbf{0}\|^2$, and $\mathbf{k} = \mathbf{h} = \mathbf{0}$, which is already included in the case $c = 0$.

Therefore, for any given $\mathbf{h}, \mathbf{k} \in \mathbb{R}^d$, if $\varphi(\mathbf{h}) = \varphi(\mathbf{k})$ and $\mathbf{h}, \mathbf{k}$ are non-negative and equimodular, we have $\mathbf{h} = \mathbf{k}$. At this point, $\varphi(\cdot)$ is injective.

## B   IMPLEMENTATION DETAILS

The experiments on SSL and UDA tasks are conducted on a NVIDIA V100 GPU, and the experiments in SHDA tasks are conducted on a NVIDIA 3090 GPU.

### B.1   SEMI-SUPERVISED LEARNING

Table 4: Parameter $\lambda$ on SSL tasks.

| Dataset | CIFAR-10 | CIFAR-100 | DTD |
|---|---|---|---|
| ERM+LRM | 0.1 | 0.5 | 0.1 |
| FlexMatch+LRM | 0.5 | 0.5 | 0.1 |
| DST+LRM | 0.1 | 0.5 | 0.1 |

The parameter $\lambda$ in Eq. (7) for SSL tasks is shown in Table 4, and the parameter $\mu$ in Eq. (7) is set to 0.1. We use mini-batch stochastic gradient descent (SGD) with a momentum of 0.9 as the optimizer, and the batch-sizes of labeled and unlabeled samples are both set to 32. Following (Chen et al., 2022), we use random-resize-crop and RandAugment (Cubuk et al., 2020) for strong augmentation and random-horizontal-filp for weak augmentation. To ensure a fair comparison, all methods utilize the same backbone for each dataset. Specifically, ResNet-18 is used on the CIFAR-10 dataset, and ResNet-50 (He et al., 2016) is adopted on the CIFAR-100 and DTD datasets. Both backbones are pretrained on ImageNet (Deng et al., 2009). Furthermore, we observe that $\mathbf{w}_c^u$ and $\mathbf{s}_c^u$ in Eq. (7) may be incorrect at the beginning of the training iteration. To prevent them from fitting into certain categories too early leading to unstable learning, we perform an additional softmax transformation before calculating cross-entropy loss, which encourages them smoother. The same strategy is also adopted in the following two tasks.

### B.2   UNSUPERVISED DOMAIN ADAPTATION

For the UDA tasks, $\lambda$ in Eq. (8) is set to 1 when used independently, and 0.1 when combined with other methods. We set the batch-sizes of both domains to 32. The optimizer is a mini-batch SGD method with a momentum of 0.9 and a learning rate annealing strategy in (Ganin et al., 2016). For a fair comparison, in all transfer tasks, we use the ResNet-50 pretrained on ImageNet as the backbone.

### B.3   SEMI-SUPERVISED HETEROGENEOUS DOMAIN ADAPTATION

For the SHDA tasks, we implement $g_s(\cdot)$ and $g_t(\cdot)$ in Eq. (9) by using a one-layer fully connected network with the Leaky ReLU (Maas et al., 2013) activation function, respectively. Analogously, we adopt a one-layer fully connected network with the softmax activation function for $f(\cdot)$ in Eq. (9). In addition, following (Yao et al., 2019), we utilize full-batch gradient descent with the Adam optimizer (Kingma & Ba, 2014) for optimizing $\{f(\cdot), g_s(\cdot), g_t(\cdot)\}$, and the learning rate is set to 0.008 when used with KPG and 0.001 in the other cases. We maintain a fixed value of $\lambda = 1$ and $\tau = 0.01$ in Eq. (9) for all the experiments. In the text-to-text scenario, The reduced dimensions of E, F, G, I, and S are 1,131, 1,230, 1,417, 1,041, and 807, respectively. Also, we randomly pick up 100 labeled articles per category as labeled source samples, while in the target domain, there are $l$ (*i.e.*, $l = 5, 10$) and 500 randomly selected labeled and unlabeled samples in each category, respectively. In the text-to-image scenario, we extract the 64-dimensional features from the fourth hidden layer of a five-layer neural network as the text features, and the 4096-dimensional $DeCAF_6$ features (Donahue et al., 2014) are extracted to represent the images in the target domain.

## C   PREDICTION DIVERSITY ANALYSIS

### C.1   ANALYSIS ON SYNTHETIC SCENARIOS

In this section, we provide detailed analyses on the prediction diversity of EntMin and LRM. In EntMin, each unlabeled sample is prone to be frequently misclassified into the dominant categories, which undermines the prediction diversity. However, the LRM estimates the label encodings through the prediction means of $C$ categories, which is calculated by the weighted average of prediction probabilities for each category. Accordingly, the LRM maintains the prediction diversity. To provide more detailed explanations, we construct two synthetic binary classification problems. One gives an explanation in the optimal situation, while the other is in the ordinary situation.

In the first example, suppose there are a total of two unlabeled samples $\mathbf{x}_1^u$ and $\mathbf{x}_2^u$, with ground-truth labels $\mathbf{y}_1^u = [1,0]$ and $\mathbf{y}_2^u = [0,1]$, respectively. We denote by $\widetilde{\mathbf{Y}_u} = \begin{bmatrix} f(g(\mathbf{x}_1^u))^{\mathrm{T}} \\ f(g(\mathbf{x}_2^u))^{\mathrm{T}} \end{bmatrix} \in \mathbb{R}^{2 \times 2}$ the prediction matrix provided by the feature extractor $g(\cdot)$ and classifier $f(\cdot)$. Next, we analyze the optimal solutions of EntMin and LRM, respectively.

(1) EntMin utilizes the soft-label of each unlabeled sample to guide its learning, and promotes sharper prediction probabilities for each unlabeled sample. Thus, the optimal solutions may be as follows:

$$\widetilde{\mathbf{Y}_u} = \begin{bmatrix} 1 & 0 \\ 1 & 0 \end{bmatrix}, \begin{bmatrix} 1 & 0 \\ 0 & 1 \end{bmatrix}, \begin{bmatrix} 0 & 1 \\ 1 & 0 \end{bmatrix}, \begin{bmatrix} 0 & 1 \\ 0 & 1 \end{bmatrix}.$$

Among those optimal solutions, the first and last solutions compromise the prediction diversity.

(2) LRM adopts the label encodings to guide the learning of prediction means. Since the label encodings for all categories are $\mathbf{e}_1 = [1,0]$ and $\mathbf{e}_2 = [0,1]$, the optimal prediction means for all categories are $\mathbf{m}_1^u = [1,0]$ and $\mathbf{m}_2^u = [0,1]$. Thus, the optimal solutions may be as follows:

$$\widetilde{\mathbf{Y}_u} = \begin{bmatrix} 1 & 0 \\ 0 & 1 \end{bmatrix}, \begin{bmatrix} 0 & 1 \\ 1 & 0 \end{bmatrix}.$$

Among those optimal solutions, all solutions maintain the prediction diversity.

In the second example, suppose there are total three unlabeled samples $\mathbf{x}_1^u$, $\mathbf{x}_2^u$ and $\mathbf{x}_3^u$, with ground-truth one-hot labels $\mathbf{y}_1^u = [1,0]$, $\mathbf{y}_2^u = [1,0]$ and $\mathbf{y}_3^u = [0,1]$, respectively. Assuming that at the $t$-th iteration, the output probabilities of the classifier for $\mathbf{x}_1^u$, $\mathbf{x}_2^u$ and $\mathbf{x}_3^u$ are $(\widetilde{\mathbf{y}}_1^u)^t = [0.7, 0.3]$, $(\widetilde{\mathbf{y}}_2^u)^t = [0.8, 0.2]$ and $(\widetilde{\mathbf{y}}_3^u)^t = [0.5, 0.5]$. Next, we detail the output probabilities that may be obtained using EntMin and LRM at the $(t+1)$-th iteration, respectively.

(1) EntMin: At the $t$-th iteration, the entropy is $\frac{1}{3}(-[0.7, 0.3] \log[0.7, 0.3]^{\mathrm{T}} - [0.8, 0.2] \log[0.8, 0.2]^{\mathrm{T}} - [0.5, 0.5] \log[0.5, 0.5]^{\mathrm{T}}) \approx 0.60$. According to the principle of EntMin, all output probabilities will become sharper and $\mathbf{x}_3$ will be pushed towards the majority category. Hence, at the $(t+1)$-th iteration, the output probabilities are highly likely to be towards $(\widetilde{\mathbf{y}}_1^u)^{t+1} = [0.8, 0.2]$, $(\widetilde{\mathbf{y}}_2^u)^{t+1} = [0.9, 0.1]$ and $(\widetilde{\mathbf{y}}_3^u)^{t+1} = [0.6, 0.4]$. At this point, although the entropy drops to $\frac{1}{3}(-[0.8, 0.2] \log[0.8, 0.2]^{\mathrm{T}} - [0.9, 0.1] \log[0.9, 0.1]^{\mathrm{T}} - [0.6, 0.4] \log[0.6, 0.4]^{\mathrm{T}}) \approx 0.50$, $\mathbf{x}_3$ is misclassified.

(2) LRM: At the $t$-th iteration, the prediction means for all categories are $(\mathbf{m}_1^u)^t = \frac{0.7[0.7, 0.3] + 0.8[0.8, 0.2] + 0.5[0.5, 0.5]}{0.7 + 0.8 + 0.5} = [0.69, 0.31]$ and $(\mathbf{m}_2^u)^t = \frac{0.3[0.7, 0.3] + 0.2[0.8, 0.2] + 0.5[0.5, 0.5]}{0.3 + 0.2 + 0.5} = [0.62, 0.38]$. The label encodings for all categories are $\mathbf{e}_1 = [1, 0]$ and $\mathbf{e}_2 = [0, 1]$. Thus, the label-encoding risk is $\frac{-[1,0] \log[0.69, 0.31]^{\mathrm{T}} - [0,1] \log[0.62, 0.38]^{\mathrm{T}}}{2} \approx 0.67$. According to the principle of LRM, the prediction means for all categories will be pushed to the label encodings. Accordingly, at the $(t+1)$-th iteration, the output probabilities are highly likely to be towards $(\widetilde{\mathbf{y}}_1^u)^{t+1} = [0.8, 0.2]$, $(\widetilde{\mathbf{y}}_2^u)^{t+1} = [0.9, 0.1]$ and $(\widetilde{\mathbf{y}}_3^u)^{t+1} = [0.4, 0.6]$. At this point, the prediction means for all categories become $(\mathbf{m}_1^u)^{t+1} = \frac{0.8[0.8, 0.2] + 0.9[0.9, 0.1] + 0.4[0.4, 0.6]}{0.8 + 0.9 + 0.4} = [0.77, 0.23]$ and $(\mathbf{m}_2^u)^{t+1} = \frac{0.2[0.8, 0.2] + 0.1[0.9, 0.1] + 0.6[0.4, 0.6]}{0.2 + 0.1 + 0.6} = [0.54, 0.46]$. Therefore, not only does the label-encoding risk decrease to $\frac{-[1,0] \log[0.77, 0.23]^{\mathrm{T}} - [0,1] \log[0.54, 0.46]^{\mathrm{T}}}{2} \approx 0.52$, but $\mathbf{x}_3$ is also classified correctly.

In summary, in the above synthetic problems, LRM always maintains the classification of all unlabeled samples into two categories, while EntMin may have several optimal solutions to classify all unlabeled samples into only one category. Accordingly, the predictions of LRM would be more diverse than those of EntMin.

## C.2 ANALYSIS ON PRACTICAL SCENARIOS

We conduct experiments to compare the prediction diversity of ERM + LRM and ERM + EntMin under class-imbalanced scenarios. To this end, we reconstruct the SHDA task of N→I into a class-imbalanced setting. As aforementioned, there are eight shared categories in both domains. Accordingly, in the source domain of N, for the first six categories, we randomly select 100 texts per category as the labeled samples. As for the last two categories, we randomly pick 20 texts as

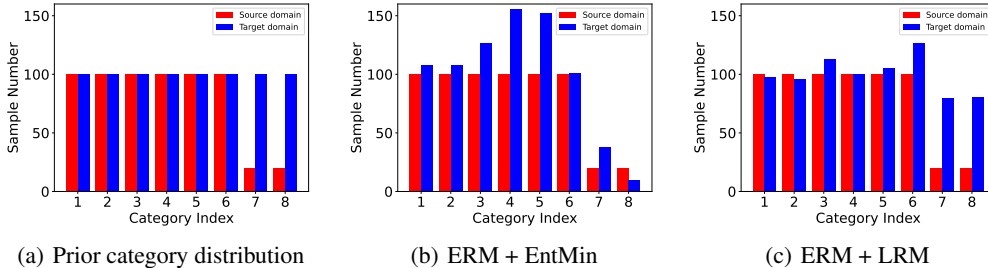

| (a) Prior category distribution | (b) ERM + EntMin | (c) ERM + LRM |

Figure 3: Empirical evaluation of prediction diversity on the SHDA task of N→I under the class-imbalanced setting. (a) The prior category distributions of the source and target domains. (b) The prior category distribution of the source domain and the category distribution of the target domain predicted by ERM + EntMin. (c) The prior category distribution of the source domain and the category distribution of the target domain predicted by ERM + LRM.

the labeled samples from each category. In the target domain of I, we randomly choose three and 100 images per category as the labeled and unlabeled samples, respectively. Figure 3(a) shows the prior category distributions of labeled samples in N and unlabeled samples in I. We can see that the prior category distribution of labeled source samples is imbalanced, whereas that of unlabeled target samples is balanced. Figure 3(b) and Figure 3(c) plot the category distributions of unlabeled samples in I predicted by ERM + EntMin and ERM + LRM, respectively. As can be seen, for the last two categories, ERM + EntMin greatly suffers from the class imbalance issue, while ERM + LRM maintains the prediction diversity well. In addition, the average classification accuracy of ERM + EntMin and ERM + LRM is 64.88% and 76.30%, respectively. Accordingly, ERM + LRM outperforms ERM+EntMin by 11.42%. One important reason is that compared to ERM + EntMin, ERM + LRM can effectively preserve prediction diversity even in class-imbalanced scenarios.

Furthermore, we also rebuild the SSL task on CIFAR-10 dataset into a class-imbalanced setting. Specifically, we randomly choose 1,000 labeled samples from each category for the first eight categories. As for the last two categories, we randomly pick 20 labeled samples per category. In the testing set, there are 1,000 samples per category. The experimental results are plotted in Figure 4. We can observe that, compared with EntMin, LRM is less susceptible to the impact of class imbalance. Moreover, the average classification accuracy of ERM + EntMin and ERM + LRM is 79.47% and 93.28%, respectively. Accordingly, ERM + LRM outperforms ERM+EntMin by a large margin of 13.81%, which verifies the effectiveness of the proposed LRM in class-imbalanced scenarios again.

In class-imbalanced supervised learning, we conduct experiments on the CIFAR-10 dataset. Specifically, in the training set, we randomly choose 50 labeled samples from each category for the last two categories, while keeping all the samples for the remaining categories. In the testing set, there are 1000 samples per category. The experimental results are shown in Figure 5. Note that the closer the model's predicted quantities for the last two categories are to 1000, the less it is affected by class imbalance. As shown in Figure 5(b), ERM + LRM yields higher predicted quantities for the last two categories compared to ERM alone, indicating that LRM can mitigate class imbalance in supervised learning. Moreover, the average classification accuracy of ERM and ERM + LRM is 87.13% and 89.41%, respectively. Accordingly, ERM + LRM outperforms ERM by 2.28%, showing the effectiveness of LRM in class-imbalanced supervised learning. Hence, compared with ERM, ERM + LRM is less susceptible to the impact of class imbalance in this dataset.

# D  MORE EXPERIMENTAL RESULTS

## D.1  RESULTS ON OFFICE-HOME DATASET FOR UDA

The results on the Office-Home dataset for UDA are listed in Table 5. As can be seen, when combined with the LRM, all the approaches achieve performance improvement. Specifically, the average classification accuracy of ERM + LRM is 69.29%, which outperforms ERM, ERM + EntMin, and ERM + BNM by 9.97%, 3.54%, and 0.58%, respectively. The results testify the superiority of LRM again.

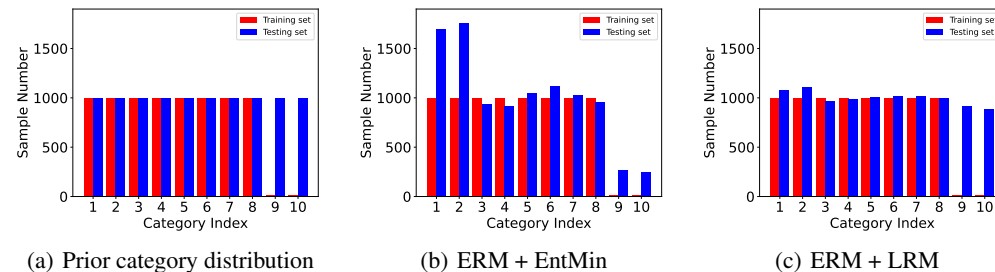

|                                  |                                  |                                  |
| :------------------------------: | :------------------------------: | :------------------------------: |
| (a) Prior category distribution  | (b) ERM + EntMin                 | (c) ERM + LRM                    |

Figure 4: Empirical evaluation of prediction diversity on the SSL task on CIFAR-10 dataset under the class-imbalanced setting. (a) The prior category distributions of the training and testing sets. (b) The prior category distribution of the training set and the category distribution of the target set predicted by ERM + EntMin. (c) The prior category distribution of the training set and the category distribution of the testing set predicted by ERM + LRM.

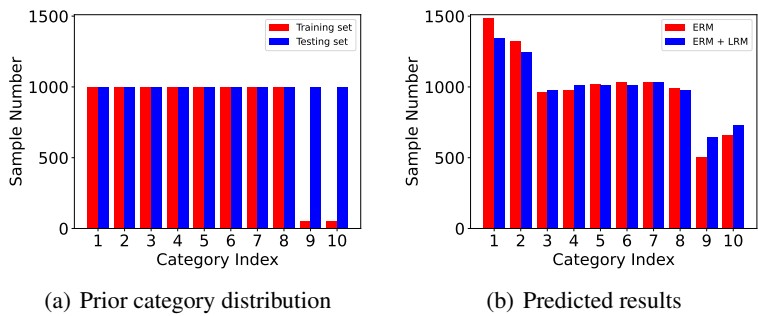

|                                  |                                  |
| :------------------------------: | :------------------------------: |
| (a) Prior category distribution  | (b) Predicted results            |

Figure 5: Empirical evaluation of prediction diversity of supervised learning on CIFAR-10 dataset under the class-imbalanced setting. (a) The prior category distributions of the training and testing sets. (b) The category distributions of the testing set predicted by ERM and ERM + LRM.

Table 5: Accuracy (%) comparison on the Office-Home dataset for UDA. The best performance of each task is marked in bold and the best performance in each comparison group is underlined.

| Method | Ar→Cl | Ar→Pr | Ar→Rw | Cl→Ar | Cl→Pr | Cl→Rw | Pr→Ar | Pr→Cl | Pr→Rw | Rw→Ar | Rw→Cl | Rw→Pr | Average |
| --- | --- | --- | --- | --- | --- | --- | --- | --- | --- | --- | --- | --- | --- |
| ERM (Vapnik, 1999) | 44.00 | 67.16 | 74.19 | 52.98 | 61.65 | 64.29 | 52.12 | 39.10 | 73.01 | 64.33 | 43.73 | 75.29 | 59.32 |
| DANN (Ganin et al., 2016) | 52.84 | 62.90 | 73.46 | 56.26 | 67.42 | 68.01 | 58.40 | 54.41 | 78.92 | 70.65 | 60.31 | 80.79 | 65.36 |
| AFN (Xu et al., 2019) | 52.68 | 72.27 | 76.96 | 65.13 | 71.13 | 72.78 | 63.93 | 51.33 | 77.81 | 72.12 | 57.52 | 81.98 | 67.97 |
| ERM+EntMin (Grandvalet & Bengio, 2004) | 46.83 | 67.28 | 77.24 | 62.23 | 70.26 | 71.69 | 59.22 | 47.06 | 79.34 | 70.92 | 54.47 | 82.43 | 65.75 |
| ERM+BNM (Cui et al., 2020) | 54.78 | 74.86 | 79.51 | 63.04 | 72.00 | 74.99 | 61.41 | 52.21 | 80.27 | 71.39 | 57.33 | 82.77 | 68.71 |
| ERM+LRM | 55.41 | 74.57 | 79.67 | 64.28 | 74.46 | 75.66 | 62.48 | 52.39 | 80.33 | 71.51 | 57.96 | 82.80 | 69.29 |
| CDAN (Long et al., 2018) | 55.18 | 72.63 | 78.01 | 62.01 | 72.46 | 73.11 | 62.68 | 53.99 | 79.65 | 72.83 | 58.23 | 83.60 | 68.70 |
| CDAN+EntMin | 54.82 | 72.43 | 78.90 | 63.03 | 72.51 | 72.87 | 62.23 | 53.53 | 80.04 | 72.42 | 58.14 | 83.73 | 68.72 |
| CDAN+BNM | 56.00 | 74.00 | 78.94 | 63.59 | 73.31 | 73.79 | 62.53 | 53.91 | 81.05 | 73.03 | 59.08 | 83.55 | 69.40 |
| CDAN+LRM | 57.08 | 74.43 | 78.84 | 64.28 | 74.09 | 74.56 | 63.70 | 54.95 | 81.15 | 73.30 | 59.46 | 84.29 | 70.01 |
| SDAT (Rangwani et al., 2022) | 57.66 | 77.06 | 81.30 | 66.07 | 76.14 | 75.91 | 63.23 | 55.92 | 81.85 | 75.87 | 62.36 | 85.41 | 71.57 |
| SDAT+EntMin | 56.97 | 77.74 | 81.33 | 65.90 | 75.83 | 75.99 | 63.58 | 55.36 | 82.30 | 75.09 | 62.01 | 85.32 | 71.45 |
| SDAT+BNM | 57.59 | 76.95 | 80.84 | 66.13 | 75.33 | 75.65 | **65.49** | 56.07 | 81.87 | 75.38 | 62.47 | 85.51 | 71.61 |
| SDAT+LRM | **58.20** | **77.75** | **82.25** | **67.04** | **76.89** | **76.96** | 64.73 | **56.46** | **82.50** | **75.87** | **63.76** | **85.97** | **72.36** |

## D.2 PARAMETER SENSITIVITY

We investigate the parameter sensitivity on the SHDA tasks of E→S5 and N→I. We mainly investigate the sensitivity of the parameter $\lambda$ in Eq. (9), as it controls the importance of the LRM. According to the accuracy w.r.t. $\lambda$ as shown in Figure 6, we can observe that when $\lambda = 0$, ERM + LRM degenerates to the original ERM without LRM. Therefore, in the initial phases of increasing the value of $\lambda$, the significant improvement in performance indicates the effectiveness of LRM. As the value of $\lambda$ increases, the performance improves and reaches an optimal value around $\lambda = 1$. Subsequently, the performance decreases gradually with the increase in the value of $\lambda$. One possible reason is that, with a large $\lambda$, the model excessively focuses on LRM with unlabeled samples and neglects ERM on labeled samples. In summary, the default setting, i.e., $\lambda = 1$, can lead to good performance on both tasks, which suggests that the default setting is a good choice for $\lambda$.

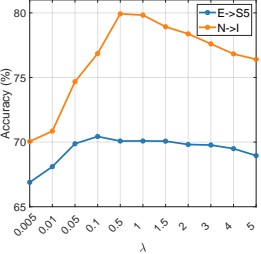

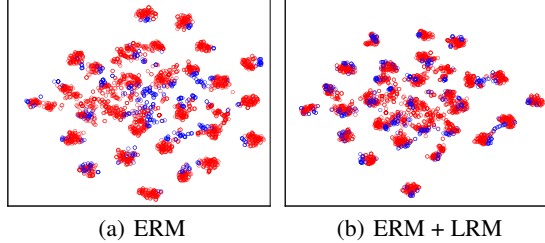

(a) ERM  (b) ERM + LRM

Figure 6: Parameter sensitivity analysis on the SHDA tasks of E → S5 and N → I.

Figure 7: t-SNE visualization for the UDA task A→D on the Office-31 dataset. The red and blue circles represent the source and target features, respectively.

Table 6: Average entropy comparison on the SSL task of four samples per category on CIFAR-10.

| Method | Entropy |
|---|---|
| ERM | 0.3832 |
| ERM + EntMin | 0.0266 |
| ERM + LRM | 0.0644 |

### D.3 FEATURE VISUALIZATION

Figure 7 visualizes the t-SNE embeddings (Van der Maaten & Hinton, 2008) of the learned source and target features on the UDA task of A→D for ERM and ERM + LRM. As can be seen, compared with the ERM alone, ERM + LRM can better align the distributions across domains. Although the LRM does not directly reduce the distributional divergence between domains, it implicitly reduces the distributional divergence across domains by aligning with label encodings, leading to a better transfer performance.

### D.4 PREDICTION PROBABILITY ANALYSIS

Table 6 shows the average entropy results of all unlabeled samples on the SSL task of four samples per category on CIFAR-10 dataset. As can be seen, ERM + EntMin and ERM + LRM obtain lower entropy than ERM, showing that both EntMin and LRM can obtain sharp prediction probabilities.

