# OpenReview forum: "Label-encoding Risk Minimization under Label Insufficient Scenarios"
_ICLR.cc/2024/Conference — Submitted to ICLR 2024_

### Official Review · Reviewer_Geja · 2023-10-27

**Soundness:** 2 fair
**Presentation:** 2 fair
**Contribution:** 2 fair
**Rating:** 5
**Confidence:** 4

**Summary:**

The authors propose a Label-encoding Risk Minimization (LRM), which draws inspiration from the phenomenon of neural collapse. Specifically, the proposed LRM first estimates the label encodings through prediction means for unlabeled samples and then aligns them with their corresponding ground-truth label encodings. As a result, the LRM takes both the prediction discriminability and diversity into account and can be utilized as a plugin in existing models to address scenarios with insufficient labels. Theoretically, the authors analyze the relationship between the LRM and ERM.

**Strengths:**

- The authors propose a new method to extend the limitation of classical Empirical Risk Minimization (ERM) into the label insufficient scenario.
- The experiments have been conducted to illustrate the superiority of the proposed method.

**Weaknesses:**

- The authors validate the performance of the proposed only on some small-scale datasets, such as CIFAR-10, CIFAR-100, an so on. The validation on larger-scale datasets are missing.
- The qualitative analysis and visualization in experiments are missing.

**Questions:**

- The authors validate the performance of the proposed only on some small-scale datasets, such as CIFAR-10, CIFAR-100, an so on. The validation on larger-scale datasets are missing.
- The qualitative analysis and visualization in experiments are missing.

---

> ### Author Response · Authors · 2023-11-22
> **Reply to Reviewer Geja**
>
> Thank you for your thoughtful review and valuable feedback. We address your concerns as follows.
> >W1. The authors validate the performance of the proposed only on some small-scale datasets, such as CIFAR-10, CIFAR-100, and so on. The validation on larger-scale datasets are missing.
>
> **AW1.** Thanks for your suggestion. Due to the time limit during the rebuttal period and the limitations of computational resources, we conduct semi-supervised learning experiments on the Tiny-ImageNet dataset [R1]. Tiny-ImageNet is a rescaled version of ImageNet-1k, which has 10,000 training images and 10,000 validation images of 200 classes. As can be seen in the following Table, LRM achieves better performance than both EntMin and BNM, which once again implies the effectiveness of LRM.
>
> | Dataset              |           | Tiny-ImageNet |           |           |
> |----------------------|-----------|---------------|-----------|-----------|
> | # Label per category | 1         |               | 4         |           |
> |                      | Top-1     | Top-5         | Top-1     | Top-5     |
> | ERM                  | 23.83     | 46.71         | 51.60     | 76.34     |
> | ERM+EntMin           | 22.04     | 44.81         | 54.61     | 81.16     |
> | ERM+BNM              | 34.88     | 58.48         | 64.00     | 85.56     |
> | ERM+LRM              | **36.99** | **59.80**     | **65.37** | **86.49** |
>
> Moreover, we conduct experiments on the UDA task, i.e., ImageNet $\rightarrow$ ImageNet-Renditions (ImageNet-R) [R2]. ImageNet-R contains 30,000 images of 200 objects in various renditions. The textures and local image statistics of ImageNet-R are different from those of ImageNet images. The experimental results are shown in the following Table. As can be seen, when combined with the LRM, all the methods achieve their best performance.
> | ImageNet   | →         | ImageNet-R  |           |             |           |
> |------------|-----------|-------------|-----------|-------------|-----------|
> | ERM        | 35.57     | CDAN        | 53.94     | SDAT        | 55.47     |
> | ERM+EntMin | 39.42     | CDAN+EntMin | 55.51     | SDAT+EntMin | 56.48     |
> | ERM+BNM    | 39.75     | CDAN+BNM    | 55.80     | SDAT+BNM    | 55.73     |
> | ERM+LRM    | **40.85** | CDAN+LRM    | **56.13** | SDAT+LRM    | **57.10** |
>
> **References**
>
> [R1] Patryk Chrabaszcz, Ilya Loshchilov, and Frank Hutter. A downsampled variant of imagenet as an alternative to the cifar datasets. *arXiv preprint arXiv:1707.08819*, 2017.
>
> [R2] D. Hendrycks, S. Basart, N. Mu, S. Kadavath, F. Wang, E. Dorundo, R. Desai, T. Zhu, S. Parajuli, M. Guo et al., “The many faces of robustness: A critical analysis of out-of-distribution generalization,” In *Proceedings of the IEEE/CVF International Conference on Computer Vision*, 2021.
>
> >W2. The qualitative analysis and visualization in experiments are missing.
>
> **AW2.** **In Appendix D.3 of our manuscript, we have conducted the feature visualization experiments**. Concretely, Figure 7 visualizes the t-SNE embeddings of the learned source and target features on the UDA task of A$\rightarrow$D for ERM and ERM + LRM. As can be seen, compared with the ERM alone, ERM + LRM can better align the distributions across domains. Although the LRM does not directly reduce the distributional divergence between domains, it implicitly reduces the distributional divergence across domains by aligning with label encodings, leading to a better transfer performance. In addition, **we have performed some quantitative analyses, i.e., the analysis of parameter sensitivity and prediction diversity**. For more details, please refer to Appendix D.2 and C in our manuscript.

---

### Official Review · Reviewer_UtTT · 2023-10-31

**Soundness:** 3 good
**Presentation:** 2 fair
**Contribution:** 2 fair
**Rating:** 5
**Confidence:** 3

**Summary:**

Empirical Risk Minimization (ERM) performs well with the sufficient labels, but suffers from neural collapse in situations with insufficient labels.  In this paper, authors prove that label encoding (one-hot encoding) is the cause of Neural Class-Mean Collapse (NCC), and propose Label-encoding Risk Minimization (LRM) to alleviate the NCC problem.

**Strengths:**

- The authors have demonstrated the cause of NCC through the use of label encoding and have further enhanced it to improve network performance.
- The evidence supporting NCC and LRM is explicitly presented.
- Experimental results show that LRM achieves good performance in various tasks such as semi-supervised learning (SSL), unsupervised domain adaptation (UDA), and semi-supervised heterogeneous domain adaptation (SHDA).

**Weaknesses:**

- Although authous conducted experiments on a variety of tasks, for each task they only ran experiments on specific datasets. For example, for UDA, they only ran experiments on Office-31 and did not evaluate on many other UDA datasets.

- Authours also do not have experimental results on large scale data. It is not confirmed that their algorithm is useful when training a model from scratch on large scale training data.

**Questions:**

- Can we say that domain adaptation is a situation where there are not enough labels? A typical domain adaptation is a setting where there are no target labels at all, and the amount of labels in the source data is irrelevant. What does it mean to have insufficient labels in this situation?

- Is there a reason why you didn't perform experiments on a large scale dataset such as ImageNet?

---

> ### Author Response · Authors · 2023-11-22
> **Reply to Reviewer UtTT (1/2)**
>
> Thank you for your thoughtful review and valuable feedback. We address your concerns as follows.
>
> >W1.
> Although authors conducted experiments on a variety of tasks, for each task they only ran experiments on specific datasets. For example, for UDA, they only ran experiments on Office-31 and did not evaluate on many other UDA datasets.
>
> **AW1.**
> Due to page limit, the experimental results of the Office-Home dataset are shown in Table 5 of Appendix D.1.
> In the manuscript, we conduct experiments on SSL, UDA, and SHDA. In the case of SSL, we evaluate the performance of LRM using the CIFAR-10, CIFAR-100, and DTD datasets. For UDA, the performance of LRM is evaluated on the Office-31 and Office-Home datasets. For SHDA, LRM is evaluated on the Multilingual Reuters Collection dataset, NUS-WIDE, and ImageNet datasets. Here, we conduct further experiments on the Tiny-ImageNet dataset for SSL, and ImageNet $\rightarrow$ ImageNet-R task for UDA.
>
> >W2.
> Authors also do not have experimental results on large scale data. It is not confirmed that their algorithm is useful when training a model from scratch on large scale training data.
>
> **AW2.** Thanks for your suggestion. Due to the time limit during the rebuttal period and the limitations of computational resources, we conduct experiments with one and four labeled samples for each category for semi-supervised learning on the Tiny-ImageNet dataset [R1]. Tiny-ImageNet is a rescaled version of ImageNet-1k, which has 10,000 training images and 10,000 validation images of 200 classes. As shown in the following table, LRM achieves better performance than both EntMin and BNM, which once again demonstrates the effectiveness of LRM.
>
> | Dataset              |           | Tiny-ImageNet |           |           |
> |----------------------|-----------|---------------|-----------|-----------|
> | # Label per category | 1         |               | 4         |           |
> |                      | Top-1     | Top-5         | Top-1     | Top-5     |
> | ERM                  | 23.83     | 46.71         | 51.60     | 76.34     |
> | ERM+EntMin           | 22.04     | 44.81         | 54.61     | 81.16     |
> | ERM+BNM              | 34.88     | 58.48         | 64.00     | 85.56     |
> | ERM+LRM              | **36.99** | **59.80**     | **65.37** | **86.49** |
>
> Moreover, we conduct experiments on the UDA task, i.e., ImageNet $\rightarrow$ ImageNet-Renditions (ImageNet-R) [R2]. ImageNet-R contains 30,000 images of 200 objects in various renditions. The textures and local image statistics of ImageNet-R are different from those of ImageNet images. The experimental results are shown in the following Table. As can be seen, when combined with the LRM, all the methods achieve their best performance.
>
> | ImageNet   | →         | ImageNet-R  |           |             |           |
> |------------|-----------|-------------|-----------|-------------|-----------|
> | ERM        | 35.57     | CDAN        | 53.94     | SDAT        | 55.47     |
> | ERM+EntMin | 39.42     | CDAN+EntMin | 55.51     | SDAT+EntMin | 56.48     |
> | ERM+BNM    | 39.75     | CDAN+BNM    | 55.80     | SDAT+BNM    | 55.73     |
> | ERM+LRM    | **40.85** | CDAN+LRM    | **56.13** | SDAT+LRM    | **57.10** |
>
> **References**
>
> [R1] Patryk Chrabaszcz, Ilya Loshchilov, and Frank Hutter. A downsampled variant of imagenet as an alternative to the cifar datasets. *arXiv preprint arXiv:1707.08819*, 2017.
>
> [R2] D. Hendrycks, S. Basart, N. Mu, S. Kadavath, F. Wang, E. Dorundo, R. Desai, T. Zhu, S. Parajuli, M. Guo et al., “The many faces of robustness: A critical analysis of out-of-distribution generalization,” In *Proceedings of the IEEE/CVF International Conference on Computer Vision*, 2021.

---

> ### Author Response · Authors · 2023-11-22
> **Reply to Reviewer UtTT (2/2)**
>
> >Q1.
> Can we say that domain adaptation is a situation where there are not enough labels? A typical domain adaptation is a setting where there are no target labels at all, and the amount of labels in the source data is irrelevant. What does it mean to have insufficient labels in this situation?
>
> **AQ1**. Yes, your understanding is right. In unsupervised domain adaptation, there is no labeled data in the target domain and hence there are not enough labeled data in the target domain. In this case, we use abundant labeled data in the source domain to help learn from unlabeled data in the target domain. This setting is consistent with the label insufficient scenario, which follows [R3] and is defined in lines 5-13 of our manuscript that "In practice, however, we often encounter some label insufficient scenarios [R3], where the labeled samples are limited or may even be absent altogether. For the former, we can utilize a large number of unlabeled samples to assist the learning of labeled samples, which falls within the scope of semi-supervised learning. On the contrary, as for the latter, a popular solution is to borrow the knowledge from a similar and label sufficient domain, i.e., source domain, for facilitating the learning of unlabeled samples, which pertains to the field of transfer learning. **The commonality of those techniques is to fully utilize unlabeled samples for improving the generalization capability in label insufficient scenarios**".
>
> **References**
>
> [R3] Shuhao Cui, Shuhui Wang, Junbao Zhuo, Liang Li, Qingming Huang, and Qi Tian. Towards discriminability and diversity: Batch nuclear-norm maximization under label insufficient situations. In *Proceedings of the IEEE/CVF Conference on Computer Vision and Pattern Recognition*, pp. 3941–3950, 2020.
>
> >Q2.
> Is there a reason why you didn't perform experiments on a large scale dataset such as ImageNet?
>
> **AQ2**.
> Please refer to AW2 for details.

---

### Official Review · Reviewer_VuKt · 2023-10-31

**Soundness:** 3 good
**Presentation:** 3 good
**Contribution:** 3 good
**Rating:** 6
**Confidence:** 5

**Summary:**

In this paper, authors focus on extending the empirical risk minimization (ERM) of the supervised learning to the label insufficient scenarios with proposed label-encoding risk minimization (LRM), which draws inspiration from the phenomenon of neural class-mean collapse (NCC). The core idea of LRM comes from estimating the label encodings through prediction means for unlabeled samples and aligning them with their corresponding ground-truth label encodings.

**Strengths:**

Specifically, it is implemented by adding LRM term to objective function. Authors not only analyze the relationship between the LRM and ERM in theory, but also demonstrate the superiority of the LRM under several label insufficient scenarios including semi-supervised learning (SSL), unsupervised domain adaptation (UDA) and semi-supervised heterogeneous domain adaptation (SHDA). LRM also can be utilized as a plugin in existing models to cope with insufficient label scenarios. Moreover, parameter sensitivity and feature visualization also be analyzed with elaborate experiments. The paper is overall well written.
Although, the calculation of the prediction means of unlabeled samples is simple, the idea of treating prediction means as an estimation of label encoding is interesting. Experimental results are attractive on several label insufficient scenarios. Fundamental proofs of theorems and implementation details in supplementary material also help with paper understanding for readers.
I'm looking forward to open source codes.

**Weaknesses:**

However, there are some questions need to be clarified from authors. Please refer to the part of questions.

**Questions:**

1.	In this paper, the core assumption is that the label encodings should remain consistent for both labeled and unlabeled samples under label insufficient scenarios. How about the performance if this assumption does not hold ? In other words, if unlabeled samples have different classes or even class is unknown (All datasets of experiment have explicit classes in manuscript, such as CIFAR, Office and so on.) how we evaluate the LRM ?
2.	Although, authors perform two tasks on class-imbalanced setting in order to verify the effectiveness of the proposed LRM. This is no theoretical guarantee on class-imbalanced supervised learning. If it exceeds the scope of this work, do we need more experiments to support the conclusion ? Because, in the real world, class-imbalanced problem is pervasive.
3.	In Figure 2(a), please give the detailed explanation why the accuracy of ERM is fluctuant while that of ERM+LRM is not. It is hard to understand.

---

> ### Author Response · Authors · 2023-11-22
> **Reply to Reviewer VuKt**
>
> Thank you for your thoughtful review and valuable feedback. We address your concerns as follows.
> >Q1. In this paper, the core assumption is that the label encodings should remain consistent for both labeled and unlabeled samples under label insufficient scenarios. How about the performance if this assumption does not hold? In other words, if unlabeled samples have different classes or even class is unknown (All datasets of experiment have explicit classes in manuscript, such as CIFAR, Office and so on.) how we evaluate the LRM?
>
> **AQ1.** The proposed LRM is based on the assumption that label encodings should remain consistent. Based on your suggestion, we conduct experiments on Partial Domain Adaptation (PDA) [R1] and Open Set Domain Adaptation (OSDA) [R2] using the Office-31 dataset. In PDA, the categories in the target domain are a subset of those in the source one. Following [R1], the source domain has 31 categories, while the target domain includes only 10 of them. Instead, in OSDA, the categories in the source domain are a subset of those in the target one. Following [R2], the source domain contains a total of 20 shared categories, and the target domain contains all the 31 categories, making the remaining 11 categories new to the source domain. The experimental results are presented in the following Tables. In OSDA, the harmonic mean of known and unknown categories is used as the performance measure. As can be seen, the proposed LRM can improve the performance of ERM in both PDA and OSDA scenarios. So when the assumption does not hold, LRM could be effective according to the above experimental results.
> | PDA    | AtoD     | AtoW     | DtoW     | WtoD      | DtoA     | WtoA      | Average   |
> |---------|-----------|-----------|-----------|------------|-----------|-----------|-----------|
> | ERM     | 87.26     | 78.64     | 97.63     | 99.36      | 87.37     | 87.89     | 89.69     |
> | ERM+LRM | **89.17** | **81.69** | **98.31** | **100.00** | **90.29** | **91.23** | **91.78** |
>
> | OSDA    | AtoD      | AtoW      | DtoW      | WtoD      | DtoA      | WtoA      | Average   |
> |---------|-----------|-----------|-----------|-----------|-----------|-----------|-----------|
> | ERM     | 70.59     | 87.46     | 89.80     | 72.79     | 68.21     | 67.80     | 76.11     |
> | ERM+LRM | **75.63** | **89.44** | **92.91** | **76.57** | **69.43** | **70.13** | **79.02** |
>
> [R1] Zhangjie Cao, Mingsheng Long, Jianmin Wang, and Michael I Jordan, “Partial transfer learning with selective adversarial networks,” In *Proceedings of the IEEE Conference on Computer Vision and Pattern Recognition*, pp. 2724–2732, 2018.
>
> [R2] Kuniaki Saito, Shohei Yamamoto, Yoshitaka Ushiku, and Tatsuya Harada. Open set domain adaptation by backpropagation. In *Proceedings of the European Conference on Computer Vision*, pp. 153–168, 2018.
>
> >Q2. Although, authors perform two tasks on class-imbalanced setting in order to verify the effectiveness of the proposed LRM. This is no theoretical guarantee on class-imbalanced supervised learning. If it exceeds the scope of this work, do we need more experiments to support the conclusion? Because, in the real world, class-imbalanced problem is pervasive.
>
> **AQ2.** Thanks for your suggestion. To further support the conclusion, we have conducted experiments involving class-imbalanced supervised learning on the CIFAR-10 dataset. Specifically, in the training set, we randomly choose $50$ labeled samples from each category within the last two categories, while keeping all the samples for the remaining categories. In the testing set, there are $1000$ samples per category. The experimental results are shown in Figure 5 of Appendix C.2. Note that the closer the model's predicted quantities for the last two categories are to 1000, the less it is affected by class imbalance. As shown in Figure 5(b), ERM + LRM yields higher predicted quantities for the last two categories compared to ERM alone, indicating that LRM can mitigate class imbalance in supervised learning. Moreover, the average classification accuracies of ERM and ERM + LRM are $87.13\\%$ and $89.41\\%$, respectively. Accordingly, ERM + LRM outperforms ERM by $2.28\\%$, showing the effectiveness of LRM in class-imbalanced supervised learning. Hence, compared with ERM, ERM + LRM is less susceptible to the impact of class imbalance in this dataset.
>
> >Q3.In Figure 2(a), please give the detailed explanation why the accuracy of ERM is fluctuant while that of ERM+LRM is not. It is hard to understand.
>
> **AQ3.** We are sorry for this confusion. In ERM of SHDA, we utilize all labeled samples from both domains to train a neural network. However, **we note that the distributions across source and target domains differ significantly. Also, the number of labeled samples in the target domain is extremely limited**. Accordingly, using ERM alone may not be able to achieve relatively stable performance on unlabeled target samples. We will add the explanation in the revision.

---

### Official Review · Reviewer_2ZQ3 · 2023-11-06

**Soundness:** 3 good
**Presentation:** 3 good
**Contribution:** 3 good
**Rating:** 6
**Confidence:** 3

**Summary:**

This paper proposes a Label-encoding Risk Minimization (LRM) for label-insufficient scenarios. The proposed LRM firstly estimates the label encodings through prediction means for unlabeled samples and then aligns them with their ground-truth label encodings. The authors theoretically analyze the relationship between LRM and ERM. The authors demonstrate the superiority of LRM under several label insufficient scenarios, including semi-supervised learning, unsupervised domain adaptation, and semi-supervised heterogeneous domain adaptation.

**Strengths:**

1.	The paper is well-written and easy to follow.
2.	The paper proves that the underlying cause of neural collapse (NCC) is the use of one-hot label encoding, and proposes label-encoding risk minimization, which minimizes the discrepancy between estimated label encodings of unlabeled samples and their corresponding label encodings. The proposed method is reasonable with a theoretical guarantee.
3.	The authors apply LRM on multiple label insufficient scenarios: semi-supervised learning, unsupervised domain adaptation and semi-supervised heterogeneous domain adaptation. The consistent performance gains validate the effectiveness and generality of LRM.

**Weaknesses:**

1.	The author primarily proves and experiments with methods in label insufficient scenarios. However, in real-world situations, many data exhibit a long-tail (class-imbalanced) distribution. Can this proposed method be applied to class-imbalanced supervised and semi-supervised scenarios?
2.	As shown in Table1, the authors combine LRM with two semi-supervised methods: FlexMatch and DST. Can the proposed method be combined with other SSL methods, such as MixMatch [1] and ReMixMatch [2]?
3.	As shown in Table2, the authors combine LRM with two UDA methods: CDAN and SDAT. Can the proposed method be combined with SOTA UDA methods [3]?

[1] MixMatch: A Holistic Approach to Semi-Supervised Learning, NeurIPS 2019.
[2] ReMixMatch: Semi-Supervised Learning with Distribution Alignment and Augmentation Anchoring, arxiv 2019.
[3] Patch-Mix Transformer for Unsupervised Domain Adaptation: A Game Perspective, CVPR 2023.

**Questions:**

See weakness for details.

---

> ### Author Response · Authors · 2023-11-22
> **Reply to Reviewer 2ZQ3**
>
> Thank you for your thoughtful review and valuable feedback. We address your concerns as follows.
> >W1. The author primarily proves and experiments with methods in label insufficient scenarios. However, in real-world situations, many data exhibit a long-tail (class-imbalanced) distribution. Can this proposed method be applied to class-imbalanced supervised and semi-supervised scenarios?
>
> **AW1.** In Appendix C.2 of our manuscript, we have conducted experiments on SHDA and SSL tasks under the class-imbalanced setting. The effectiveness of the proposed LRM in class-imbalanced scenarios can be verified.
>
> Here, we conduct further experiments of class-imbalanced supervised learning on the CIFAR-10 dataset. Specifically, in the training set, we randomly choose $50$ labeled samples from each category for the last two categories, while keeping all the samples for the remaining categories. In the testing set, there are $1000$ samples per category. The experimental results are shown in Figure 5 of Appendix C.2. Note that the closer the model's predicted quantities for the last two categories are to 1000, the less it is affected by class imbalance. As shown in Figure 5(b), ERM + LRM yields higher predicted quantities for the last two categories compared to ERM alone, indicating that LRM can mitigate class imbalance in supervised learning. Moreover, the average classification accuracies of ERM and ERM + LRM are $87.13\\%$ and $89.41\\%$, respectively. Accordingly, ERM + LRM outperforms ERM by $2.28\\%$, showing the effectiveness of LRM in class-imbalanced supervised learning. Hence, compared with ERM, ERM + LRM is less susceptible to the impact of class imbalance in this dataset.
>
> >W2. As shown in Table1, the authors combine LRM with two semi-supervised methods: FlexMatch and DST. Can the proposed method be combined with other SSL methods, such as MixMatch [1] and ReMixMatch [2]?
>
> **AW2.** Thanks for your suggestion. We conduct experiments with one and four labeled samples for each category for SSL on the CIFAR-100 dataset. The experimental results of MixMatch + LRM and ReMixMatch + LRM are listed as follows. As can be seen, LRM is still effective when combined with MixMatch and ReMixMatch.
> | Dataset              | CIFAR-100 |        |
> |----------------------|-----------|--------|
> | # Label per category | 1         | 4      |
> | MixMatch             | 44.08    | 74.50 |
> | MixMatch+EntMin      | 25.00     | 75.27 |
> | MixMatch+BNM         | 63.63    | 79.98 |
> | MixMatch+LRM         | 66.64    | 81.35 |
> | ReMixMatch           | 69.32    | 83.09 |
> | ReMixMatch+EntMin    | 69.39    | 83.13 |
> | ReMixMatch+BNM       | 66.67    | 83.14 |
> | ReMixMatch+LRM       | **73.45**    | **83.25** |
>
> >W3. As shown in Table2, the authors combine LRM with two UDA methods: CDAN and SDAT. Can the proposed method be combined with SOTA UDA methods [3]?
>
> **AW3.** Thanks for your suggestion. We conduct experiments for UDA on the Office-Home dataset. The experimental results of PMTrans [3] + LRM are listed as follows. As can be seen, the LRM is still effective when combined with PMTrans, outperforming both EntMin and BNM.
> | Method         | ArtoCl | ArtoPr | ArtoRw | CltoAr | CltoPr | CltoRw | PrtoAr | PrtoCl | PrtoRw | RwtoAr | RwtoCl | RwtoPr | Average |
> |----------------|--------|--------|--------|--------|--------|--------|--------|--------|--------|--------|--------|--------|---------|
> | PMTrans        | 82.05  | **92.64**  | 92.37  | 89.25  | 92.23  | 92.76  | 88.25  | 80.17  | 92.97  | 89.71  | 81.73  | 94.41  | 89.05   |
> | PMTrans+EntMin | 81.94  | 91.53  | 92.72  | 89.25  | 92.50  | 92.53  | 87.13  | 79.60  | 93.36  | 88.96  | 80.03  | 94.32  | 88.66   |
> | PMTrans+BNM    | 82.08  | 92.21  | **92.85**  | 89.42  | **92.62**  | 93.11  | 88.13  | 80.72  | 93.34  | 89.54  | 81.82  | 94.27  | 89.18   |
> | PMTrans+LRM    | **82.49**  | 92.48  | 92.74  | **89.67**  | 92.50  | **93.34**  | **88.46**  | **82.03**  | **93.45**  | **90.67**  | **83.78**  | **94.68**  | **89.69**   |

---

### Official Review · Reviewer_yg8U · 2023-11-11

**Soundness:** 1 poor
**Presentation:** 1 poor
**Contribution:** 3 good
**Rating:** 3
**Confidence:** 3

**Summary:**

The paper investigates scenarios with limited label information, such as semi-supervised learning or unsupervised domain adaptation for multiclass problems.
It proposes a new way to incorporate the unlabeled data into the risk formulation, by estimating and optimizing the mean prediction for each  class. The method is motivated by NCC, and is demonstrated to be applicable to several base methods, task types, and datasets.

**Strengths:**

* The proposed modification gives substantial improvements in the experiments, and can be combined with several different base methods.
* The proposal can be applied in several related, but distinct, settings with limited label information.
* I think the single-sample DST+LRM results for Cifar10 (~68%) are quite impressive.

**Weaknesses:**

### Claims
The paper makes several claims that are wrong or (in my opinion) overstated:


> We prove that the underlying cause of NCC roots in the use of label encoding, i.e., one-hot label encoding, which
leads to the collapse of the features through back-propagation.

I don't think this is true. On one hand, it requires the injectivity assumption made later in the paper. On the other, I believe that if you assume this, then there is no need for one-hot encodings, e.g., if you use squared error, and map each category to a distinct point in R^m, then optimizing the loss to zero requires the features to collapse for each category.

---

>  Since these label encodings serve as accurate supervision information

This doesn't seem to be true, at least not without additional qualifiers. From the appendix:

> Furthermore, we observe that $w_c^u$ and $s^u_c$ in Eq. (7) may be incorrect at the beginning of the training iteration.
> To prevent them from fitting into certain categories too early leading to unstable learning, we perform an additional softmax
> transformation before calculating cross-entropy loss, which encourages them smoother.
> The same strategy is also adopted in the following two tasks.

---

In the intro,
> One problem of EntMin is that the soft-labels assigned by the classifier could be mainly from dominant
> categories with large numbers of samples, resulting in a decrease in prediction diversity (Cui et al., 2020) that the samples are prone to be
> pushed towards the majority categories. One reason for that
> lies in the absence of more appropriate guidance information for unlabeled samples. So we want to
> ask “for unlabeled samples, is there more precise guidance information available?”

To me, this insinuates that the paper would look at imbalanced settings, as these seem to motivate this work here. In fact, though, this is deferred to future work:

> As a future direction, we intend to investigate the relationship between the LRM and ERM, in the context of
> class-imbalanced supervised learning.

---

Theorem 1:
> In addition, Theorem 1 is loss-agnostic and solely relies on the mapping property of f (·)

It is *not* loss agnostic. In fact, the theorem itself states that it assumes L(x,y) =  0 => x = y, which is not true, e.g., for max-margin losses.

> The linear classifier is an injective function, which thus satisfies Theorem 1.

A linear classifier need not be injective. In fact, if C < d, it *cannot* be injective.
Regarding softmax, if you use `f` as the softmax function, and `L` as cross-entropy, then I would no longer call `h` the "features" -- these are now the logits. And I don't think applying L2 normalization or ReLU nonlinearities to the logits is a common practice.

---

sec. 4.2
> These label encodings serve as reliable supervised information for learning from labeled samples.
> Moreover, note that the label encodings remain consistent for both labeled and unlabeled samples
> under label insufficient scenarios studied in this paper. Consequently, it is reasonable to apply
> label encodings to guide the learning process of unlabeled samples.

that is an unsubstantiated claim (at this point in the paper at least)

> Specifically, we first calculate the weighted average of prediction probabilities for unlabeled samples in each category, i.e., prediction mean.

The way this is written is wrong, though the actual calculations presented in the paper seem to be sensible. You cannot average predictions for unlabeled samples *of a category*, precisely because the categories are unknown. So the averaging is actually over the predicted distribution of categories as provided by some classifier.

### Theorem 3:
Theorem 3 seems to be the wrong way around, in the sense that normally, you'd like the original loss/risk to be upper-bounded by your proposed surrogate, so that optimizing the latter gives some guarantees on the former.

On a positive note, though, Fig. 2 shows that in practice, the two values can be quite close, so this might be more a theoretical concern.

### Readability:
The paper would benefit from grammar improvements. Mostly, these do not impact readability too much. Below are two examples where I don't think the overall sentence structure works.


page  2:
> As the ERM heavily relies on the guidance of the label information and fails to fully utilize the potential of unlabeled
samples, leading to suboptimal performance in label insufficient scenarios.

page 9:
> Accordingly, under the setting where only labeled samples from both domains are
> available for training, which can be regarded as a class-balanced supervised learning task.

**Questions:**

The label-encoding risk is a *global* quantity, in the sense that is needs to know all training points to be calculated, and does not decompose into a sum over individual points. How does that integrate into mini-batch training?

---

> [...] while in contrast, according to properties (4) and (5) in Theorem 2, LRM minimizes the label-encoding risk, which is verified in Appendix D.4.

I don't understand this argument. Doesn't the LRM minimize the Label-encoding risk by construction? What does this have to do with (4) and (5)?

---

> ### Author Response · Authors · 2023-11-22
> **Reply to Reviewer yg8U (1/3)**
>
> Thank you for your thoughtful review and valuable feedback. We address your concerns as follows.
> > W1. The paper makes several claims that are wrong or (in my opinion) overstated:
> "We prove that the underlying cause of NCC roots in the use of label encoding, i.e., one-hot label encoding, which leads to the collapse of the features through back-propagation." I don't think this is true. On one hand, it requires the injectivity assumption made later in the paper. On the other, I believe that if you assume this, then there is no need for one-hot encodings, e.g., if you use squared error, and map each category to a distinct point in $R^m$, then optimizing the loss to zero requires the features to collapse for each category.
>
> **AW1.** We appreciate your careful review. Indeed, via the NCC phenomenon, we observe that in ERM, solely $C$ ($C$ is the total number of categories) label encodings guide the learning process of labeled samples. Also, we theoretically prove that the utilization of label encodings can induce the NCC phenomenon under specific conditions. **Building upon this observation, we further find that label encodings can also serve as guidance information for unlabeled samples**. Inspired by this, we propose the LRM. **We will carefully revise the presentation in the revision**.
>
> > W2. "Since these label encodings serve as accurate supervision information." This doesn't seem to be true, at least not without additional qualifiers.
> From the appendix:
> "Furthermore, we observe that  ${\mathbf{w}}_c^u$ and ${\mathbf{s}}_c^u$ in Eq. (7) may be incorrect at the beginning of the training iteration. To prevent them from fitting into certain categories too early leading to unstable learning, we perform an additional softmax transformation before calculating cross-entropy loss, which encourages them smoother. The same strategy is also adopted in the following two tasks."
>
> **AW2.** We feel that there may be a misunderstanding about the label encoding. In the claim that "Since these label encodings serve as accurate supervision information", these label encodings are referred to **ground-truth** label encodings, i.e., $\\{\mathbf{e}_c\\}\_{c =1}^C$, in our manuscript. Here, $\mathbf{e}_c$ denotes the one-hot label encoding associated with category $c$, which is a one-hot vector where its $c$-th element equals 1 while all other elements are 0. However, in the appendix, $\mathbf{w}_c^u$ ($\mathbf{s}_c^u$) in Eq.(7) is refer to an **estimated** label encoding. It may be incorrect at the beginning of the training iteration. Thus, **the claim that "Since these label encodings serve as accurate supervision information" is correct**.
>
> > W3. In the intro,
> "One problem of EntMin is that the soft-labels assigned by the classifier could be mainly from dominant categories with large numbers of samples, resulting in a decrease in prediction diversity (Cui et al., 2020) that the samples are prone to be pushed towards the majority categories. One reason for that lies in the absence of more appropriate guidance information for unlabeled samples. So we want to ask "for unlabeled samples, is there more precise guidance information available?""
> To me, this insinuates that the paper would look at imbalanced settings, as these seem to motivate this work here. In fact, though, this is deferred to future work:
> "As a future direction, we intend to investigate the relationship between the LRM and ERM, in the context of class-imbalanced supervised learning."
>
> **AW3.** Firstly, **we want to emphasize that the motivation of this paper is to find more precise guidance information for unlabeled samples**. Based on the NCC phenomenon, we find that the guidance information of the ERM is the label encodings. Also, under label insufficient scenarios studied in this paper, the label encodings of labeled samples remain consistent with those of unlabeled samples. Accordingly, it is reasonable to apply label encodings as guidance information to guide the learning of unlabeled samples. Inspired by this, we propose the LRM. Secondly, unlike EntMin, the LRM can alleviate the **class-imbalanced issue in label insufficient scenarios (but not supervised learning)** to some extent. **In Appendix C of our manuscript, we have provided a detailed analysis and empirical comparison in label insufficient scenarios**. Thirdly, the LRM draws inspiration from NCC, which is a widely observed phenomenon within the ERM framework. Thus, **we reveal the theoretical relationship between LRM and ERM (but not EntMin) under the setting of class-balanced supervised learning (but not label insufficient scenarios)**.

---

> ### Author Response · Authors · 2023-11-22
> **Reply to Reviewer yg8U (2/3)**
>
> >W4. Theorem 1:
> "In addition, Theorem 1 is loss-agnostic and solely relies on the mapping property of $f(\cdot)$"
> It is not loss agnostic. In fact, the theorem itself states that it assumes L(x,y) = 0 => x = y, which is not true, e.g., for max-margin losses.
>
> **AW4.** We appreciate your pointing out the inappropriate statements, and we sincerely apologize for this. **We will remove inappropriate statements and carefully revise the presentation in the revision**.
>
> > W5. "The linear classifier is an injective function, which thus satisfies Theorem 1."A linear classifier need not be injective. In fact, if $C < d$, it cannot be injective. Regarding softmax, if you use f as the softmax function, and L as cross-entropy, then I would no longer call h the "features" -- these are now the logits. And I don't think applying L2 normalization or ReLU nonlinearities to the logits is a common practice.
>
> **AW5.** We agree with you that the linear classifier is an injective function only under certain conditions and under certain conditions on the input, the softmax function is injective. Here we just discuss some cases where Theorem 1 can be applied. We will revise the discussion in the revision.
>
> > W6. "These label encodings serve as reliable supervised information for learning from labeled samples. Moreover, note that the label encodings remain consistent for both labeled and unlabeled samples under label insufficient scenarios studied in this paper. Consequently, it is reasonable to apply label encodings to guide the learning process of unlabeled samples." that is an unsubstantiated claim (at this point in the paper at least)
>
> **AW6.** We want to emphasize once again that **these label encodings** refer to **ground-truth** label encodings rather than **estimated** ones. Please refer to AW2 for details. Furthermore, as an example, let's assume that there are two categories, i.e., cat, and dog, and their ground-truth (one-hot) label encodings are denoted as $[1, 0]$ and $[0,1]$, respectively. For the labeled samples, we can directly utilize those ground-truth label encodings to guide their learning. As for the unlabeled samples, based on the property (3) of Theorem 2 in our manuscript, we can first estimate the label encodings through prediction means for unlabeled samples, and then align them with their corresponding ground-truth label encodings. By doing so, the learning of unlabeled samples can be effectively guided. Accordingly, these label encodings can serve as supervision information for both labeled and unlabeled samples.
>
> > W7. "Specifically, we first calculate the weighted average of prediction probabilities for unlabeled samples in each category, i.e., prediction mean." The way this is written is wrong, though the actual calculations presented in the paper seem to be sensible. You cannot average predictions for unlabeled samples of a category, precisely because the categories are unknown. So the averaging is actually over the predicted distribution of categories as provided by some classifier.
>
> **AW7.** We are sorry for this confusion. Indeed, **for a category, we can obtain the predicted probabilities of all unlabeled samples belonging to that category, which are provided by some classifier. Then, using those predicted probabilities as weights, we can calculate the weighted average of all unlabeled samples, i.e., prediction mean**. In the revision, we will modify "Specifically, we first calculate the weighted average of prediction probabilities for unlabeled samples in each category, i.e., prediction mean. Based on the properties of prediction means, they can serve as an estimation for label encodings." to "Specifically, for a category, we first obtain the predicted probabilities via some classifier of all unlabeled samples belonging to that category. Then, using those predicted probabilities as weights, we can calculate the weighted average of all unlabeled samples, i.e., prediction mean. Based on the properties of the prediction mean, it can serve as an estimation of the label encoding corresponding to that category."
>
> > W8. Theorem 3 seems to be the wrong way around, in the sense that normally, you'd like the original loss/risk to be upper-bounded by your proposed surrogate, so that optimizing the latter gives some guarantees on the former. On a positive note, though, Fig. 2 shows that in practice, the two values can be quite close, so this might be more a theoretical concern.
>
> **AW8.** We feel that there may be a misunderstanding about Theorem 3. Theorem 3 is just to reveal the relationship between LRM and ERM but not treat LRM as a surrogate of ERM, since ERM and LRM are for different purposes that LRM is proposed to utilize unlabeled data and ERM is for labeled data. We will make it clearer in the revision.

---

> ### Author Response · Authors · 2023-11-22
> **Reply to Reviewer yg8U (3/3)**
>
> >W9. The paper would benefit from grammar improvements. Mostly, these do not impact readability too much. Below are two examples where I don't think the overall sentence structure works. (i) As the ERM heavily relies on the guidance of the label information and fails to fully utilize the potential of unlabeled samples, leading to suboptimal performance in label insufficient scenarios. (ii) Accordingly, under the setting where only labeled samples from both domains are available for training, which can be regarded as a class-balanced supervised learning task.
>
> **AW9.** Thanks for catching those for us, and we refine them as follows: (i) The ERM heavily relies on the guidance of label information, which thus fails to fully mine the potential of unlabeled samples. Accordingly, it cannot achieve good performance in label insufficient scenarios. (ii) Accordingly, when we solely utilize labeled samples from both domains for training, it can be regarded as a class-balanced supervised learning task. We will carefully revise the presentation in the revision.
>
> > Q1. The label-encoding risk is a global quantity, in the sense that is needs to know all training points to be calculated, and does not decompose into a sum over individual points. How does that integrate into mini-batch training?
>
> **AQ1.** We are sorry for this confusion. In each mini-batch, we can obtain the predicted probabilities that each sample in the mini-batch belongs to all categories, which are offered by some classifier. Then, the label encoding associated with each category can be estimated by calculating its corresponding prediction mean. Accordingly, **we can calculate the label-encoding risk in each mini-batch, which is an approximation of the label-encoding risk by using all the data**. Empirically, we find that using this simple approximation can achieve good performance. We will add such discussions in the revision.
>
> > Q2. [...] while in contrast, according to properties (4) and (5) in Theorem 2, LRM minimizes the label-encoding risk, which is verified in Appendix D.4. I don't understand this argument. Doesn't the LRM minimize the Label-encoding risk by construction? What does this have to do with (4) and (5)
>
> **AQ2.** Here we want to express that **LRM can force the prediction probability of each unlabeled sample to be sharper (i.e., close to one-hot vector) by minimizing the label-encoding risk, thereby enhancing the prediction discriminability. The properties (4) and (5) in Theorem 2 theoretically explain why minimizing the label-encoding risk can achieve this property**. This property is similar to EntMin but in different ways where EntMin achieves this property by minimizing the entropy of the prediction probability of each unlabeled sample. We conduct experiments in Appendix D.4 to verify that LRM can achieve this property even better than EntMin.
> We will make it clearer in the revision.

---

### Comment · Area_Chair_Y2yS · 2023-11-22
**Author-Reviewer Discussion ends soon**

Dear Authors! The discussion phase ends soon. This is your last chance to send your rebuttal.

Dear Reviewers! If the Authors will send their rebuttal, please try to respond to it (at least by acknowledging that you have read it).

Thank you!

Best, AC for Paper #5287

---

### Meta-Review · Area_Chair_Y2yS · 2023-12-14

**Metareview:**

The paper introduces a new algorithmic solution for a problem of learning with insufficient labels.

In general, the reviewers appreciate the presented idea and find the empirical results very promising. Unfortunately, the theoretical part of the paper contains several serious flaws, which are not in line with the expected standards for papers published at top machine learning conferences. The response given by the authors in the rebuttal has not addressed the core of the issue.

We encourage the authors to revise the paper accordingly and after succesful revision to resubmit to another top conference.

**Justification For Why Not Higher Score:**

The authors have not been able to correct the theoretical issues found by one of the reviewers, who has kept his critical opinion on the theoretical part of the paper.

**Justification For Why Not Lower Score:**

N/A

---

### Decision · Program_Chairs · 2024-01-16

Reject